# Algorithmic Recourse of In-Context Learning for Tabular Data

**Wenshuo Dong** [* 1 2 3]  **Jiaming Zhang** [* 1 4]  **Shaopeng Fu** [1]  **Hongbin Lin** [5]  **Di Wang** [1]  **Lijie Hu** [2]

## Abstract

As predictive models are increasingly deployed in high-stakes settings such as credit approval, there is a growing need for post-hoc methods that provide recourse to affected individuals. Many such models operate on tabular data, where features correspond to real-world attributes. Recently, in-context learning (ICL) has enabled large language models to perform tabular prediction by conditioning on labeled examples at inference time, without explicit training. However, algorithmic recourse for tabular decision-making under ICL remains largely unexplored. In this work, we present the first study of algorithmic recourse for tabular data under ICL. We carry out a theoretical analysis, showing that recourse remains well-defined and bounded, and we characterize how recourse converges toward classical solutions as the context size increases. In practice, we propose a novel zeroth-order recourse framework, Adaptive Subspace Recourse for In-Context Learning (ASR-ICL), that efficiently generates actionable and sparse recourse for black-box ICL models. The proposed framework naturally extends to multiclass tabular tasks. Experiments across multiple real-world datasets and models demonstrate that ASR-ICL achieves recourse quality comparable to existing methods with fewer queries and empirically confirm the predicted convergence behavior, supporting our theoretical analysis.

## 1. Introduction

Machine learning models are increasingly deployed in high-stakes decision-making domains such as credit lending and criminal justice (Barocas & Selbst, 2016). Consequently, there is a growing emphasis on providing algorithmic recourse to individuals who are adversely impacted by predictions, offering actionable guidance on how to obtain a favorable outcome (Voigt & Von dem Bussche, 2017). For example, when a loan application is rejected by an automated decision system, algorithmic recourse can suggest feasible actions, such as improving income stability or reducing outstanding debt, that may lead to a different outcome. Algorithmic recourse is an explanation framework that provides individuals with actionable and feasible feature changes that could lead to a favorable decision outcome (Ustun et al., 2019). Unlike post-hoc explanations that describe why a decision was made, recourse focuses on actionable changes that an individual can implement to obtain a different outcome (Wachter et al., 2017). As a result, algorithmic recourse has become a central component of responsible and trustworthy machine learning.

Many high-stakes decision systems operate on tabular data, where features correspond to semantically meaningful and potentially actionable real-world attributes such as income, employment status, or medical indicators (Kleinberg et al., 2018; Pawelczyk et al., 2020). This makes tabular prediction an important setting for algorithmic recourse since recourse feature changes can often be interpreted as feasible actions. Recently, in-context learning (ICL) with large language models (LLMs) has emerged as a powerful paradigm for tabular prediction (Wen et al., 2025). Models such as TabPFN and TabICL demonstrate that strong tabular predictors can be obtained without explicit training by conditioning on a set of labeled examples provided at inference time (Hollmann et al., 2025; Qu et al., 2025). In this paradigm, predictions are dynamically induced from the context rather than produced by a fixed trained model. As a result, the effective decision rule varies across contexts, rendering the predictor inherently black-box and input-dependent.

Many prior works have successfully applied algorithmic recourse to trained predictive models using various optimization techniques, such as gradient-based methods, integer programming, and causal modeling (Karimi et al., 2021; Ustun et al., 2019). These approaches typically assume a fixed decision function obtained after training, under which recourse is defined and computed (Mothilal et al., 2020; Ustun et al., 2019; Poyiadzi et al., 2020; Pawelczyk et al., 2020).

---
[*]Equal contribution  [1]King Abdullah University of Science and Technology, Saudi Arabia [2]Mohamed bin Zayed University of Artificial Intelligence, United Arab Emirates [3]University of Copenhagen, Denmark [4]Renmin University of China, China [5]The Hong Kong University of Science and Technology(Guangzhou), China. Correspondence to: Di Wang <di.wang@kaust.edu.sa>.

*Proceedings of the 43rd International Conference on Machine Learning*, Seoul, South Korea. PMLR 306, 2026. Copyright 2026 by the author(s).

In contrast, in-context learning induces predictions dynamically from the provided context; thus, the effective decision rule can vary across contexts even for the same individual (Brown et al., 2020). The effective models faced by the user is context-conditioned: changing the context can change both the prediction and the recourse action, even for the same query instance. Therefore, recourse under ICL raises new questions that do not arise in the classical fixed-model setting, including whether recourse is well-defined for a given context, whether it remains stable across contexts, and whether it can be computed efficiently with only black-box access. Despite the growing adoption of ICL for tabular prediction, the implications of such dynamic decision rules for algorithmic recourse remain largely unexplored.

In this work, we present the first systematic study of algorithmic recourse under in-context learning for tabular data, addressing a critical gap between recent advances in tabular ICL and the absence of recourse in such settings. On the theoretical side, we formalize recourse with respect to context-conditioned predictors and establish that recourse remains well-defined and admits feasibility and boundedness guarantees. In particular, for a linear regression task trained via a single-layer linear self-attention model, we show that the cost of optimal recourse can be upper bounded and that these bounds tighten as the number of context examples increases, revealing a convergence of ICL-based recourse toward classical recourse for trained models. In practice, we further propose Adaptive Subspace Recourse for In-Context Learning (ASR-ICL), an efficient zeroth-order recourse framework for black-box ICL models. ASR-ICL adaptively concentrates the search on a small subset of important features, enabling efficient recourse computation in high-dimensional tabular spaces without access to gradients or model internals. Importantly, ASR-ICL naturally extends beyond binary classification and supports multi-class tabular decision tasks without task-specific modifications. Extensive experiments on multiple real-world datasets and across a range of tabular ICL models and general-purpose language models show that ASR-ICL achieves recourse validity comparable to existing methods for trained models while consistently yielding lower recourse costs. We further empirically confirm the theoretical convergence behavior as the number of context instances increases and demonstrate that ASR-ICL naturally extends to multi-class tabular decision tasks.

Our main contributions are summarized as follows:

- We present the first theoretical analysis of algorithmic recourse under in-context learning, establishing that recourse remains well-defined and bounded despite the dynamic, context-conditioned nature of ICL. We further show that, as the number of context examples increases, ICL-based recourse converges to classi-

cal linear-model recourse with provable perturbation bounds.

- In practice, we propose ASR-ICL, the first recourse framework specifically designed for black-box ICL models, which enables efficient and sparse recourse in high-dimensional spaces under expensive model queries via adaptive zeroth-order optimization.

- Extensive experiments on diverse real-world datasets and across multiple models demonstrate that ASR-ICL achieves recourse validity and cost comparable to existing methods for trained models, while naturally supporting multi-class tabular decision tasks. In particular, our results confirm the predicted stabilization of recourse behavior as the number of context instances increases.

## 2. Related Work

**Algorithmic Recourse.** Prior works have studied algorithmic recourse for predictive models, aiming to generate counterfactual explanations that suggest actionable changes to reverse unfavorable decisions, primarily in binary classification settings (Karimi et al., 2020; De Toni et al., 2025; Gao & Lakkaraju, 2023; Kanamori et al., 2024; Ross et al., 2021). Beyond classification, CARAT (Datta et al., 2022) extends recourse to unsupervised anomaly detection in tabular data. These methods are typically developed for trained models with a fixed decision function, under which recourse is defined and computed. Representative approaches include gradient-based optimization methods, integer programming formulations, and black-box search techniques for tabular data (Wachter et al., 2017; Ustun et al., 2019; Poyiadzi et al., 2020; Pawelczyk et al., 2020). Several extensions incorporate sparsity, causal or manifold constraints, diversity, and robustness considerations (Karimi et al., 2021; Upadhyay et al., 2021; Turbal et al., 2025; Ehyaei et al., 2023). Importantly, despite these extensions, existing approaches continue to assume a fixed decision function obtained after training (Kanamori et al., 2026). In contrast, in-context learning induces a predictor that is constructed dynamically from the context examples provided at inference time, causing the effective decision rule to vary across contexts even for the same individual. This setting violates the core assumption underlying existing recourse formulations and raises fundamental questions about the well-definedness, stability, and computability of recourse. To the best of our knowledge, algorithmic recourse under in-context learning has not been previously studied.

**ICL for Tabular Data** Recent work has demonstrated that ICL can be highly effective for tabular prediction (Thomas et al., 2024; Carey et al., 2024; Breejen et al., 2024; Küken et al., 2025; Wen et al., 2025; Hollmann et al., 2025; Grin-

sztajn et al., 2025). Existing approaches broadly fall into two categories. The TabPFN family employs numerical representations and is trained on large collections of synthetic tabular tasks, yielding strong performance as a specialized in-context predictor for tabular data (Hollmann et al., 2023; 2025). In parallel, LLM-based approaches reformulate tabular inputs as text and leverage general-purpose language models, enabling flexible integration with standard LLM infrastructures and downstream applications (Wen et al., 2025; Qu et al., 2025; Gardner et al.). Despite these differences, both works induce predictions dynamically from context examples at inference time, making the effective decision rule context-dependent rather than fixed after training. Most existing studies on tabular ICL focus on accuracy, scalability, and representation design. Relatedly, TABGEN-ICL (Fang et al., 2025) focuses on in-context learning for tabular data synthesis rather than recourse generation. However, algorithmic recourse for tabular ICL models has not yet been investigated, despite its importance in high-stakes decision-making.

## 3. Preliminaries

Let $x \in \mathcal{X}$ denote an instance with $d$ features, where $\mathcal{X} \subseteq \mathbb{R}^{d_c} \times \mathcal{Z}^{d_d}$ consists of $d_c$ continuous and $d_d$ discrete attributes. Let $f : \mathcal{X} \to \mathcal{Y}$ be a fixed predictive model, where $\mathcal{Y} = \{0, 1\}$ for binary classification or $\mathcal{Y} = \{1, \ldots, C\}$ for multi-class classification. Given an input $x$, the model outputs a prediction $y = f(x)$. We focus on the scenario where the model receives an unfavorable outcome and the desired target label is $y^+ \in \mathcal{Y}$.

**Algorithmic Recourse.** Algorithmic recourse aims to provide actionable modifications that would change an unfavorable prediction to a desired one. Let $x' \in \mathcal{X}$ denote a candidate recourse for $x$. A valid recourse satisfies $f(x') = y^+$. Among all such candidates, recourse seeks the minimal-cost modification under a cost function $c : \mathcal{X} \times \mathcal{X} \to \mathbb{R}_{\geq 0}$, subject to feasibility constraints $\Omega(x)$ that encode feature-level actionability (e.g., immutability and monotonicity):

$$x^* = \arg\min_{x' \in \Omega(x)} c(x, x') \quad \text{s.t.} \quad f(x') = y^+. \quad (1)$$

**In-context Learning.** In-context learning (ICL) refers to the prediction paradigm in which a model performs a task by conditioning on a set of labeled examples provided at inference time, without explicit parameter updates.

Let $\mathcal{C} = \{(x_i, y_i)\}_{i=1}^k$ denote a context set of $k$ labeled examples, where $x_i \in \mathcal{X}$ and $y_i \in \mathcal{Y}$. Given a query instance $x \in \mathcal{X}$, an in-context learner induces a predictor

$$f_\mathcal{C}(x) = \text{LLM}(\mathcal{C}, x),$$

where $\text{LLM}(\cdot)$ denotes a large language model evaluated on a prompt containing the context examples and the query instance.

Unlike conventional predictors $f$ that are fixed after training, the in-context predictor $f_\mathcal{C}$ is not fixed but depends on the choice and composition of the context set $\mathcal{C}$. This context-conditioned and implicit decision rule fundamentally differs from standard trained models and forms the basis of our study of algorithmic recourse for ICL.

## 4. Theoretical Analysis

In this section, we establish the theoretical foundations of algorithmic recourse within the in-context learning framework. We first formalize the problem of algorithmic recourse under the ICL theory, then investigate the conditions for recourse feasibility, and finally derive a high-probability upper bound for the optimal perturbation cost.

### 4.1. Problem Setup

**In-context Predictor.** Following the setup in Zhang et al. (2024), we consider a linear regression task where each instance is defined by a task weight $w \sim \mathcal{N}(0, I_d)$ and covariates $x_i \sim \mathcal{N}(0, \Lambda)$ with a positive definite covariance matrix $\Lambda \in \mathbb{R}^{d \times d}$. The labels are generated as $y_i = \langle w, x_i \rangle$. Given a test prompt of length $M$, $P = (x_1, y_1, \ldots, x_M, y_M, x_{\text{query}})$, the data is organized into an embedding matrix $E \in \mathbb{R}^{(d+1) \times (M+1)}$:

$$E = \begin{pmatrix} x_1 & x_2 & \ldots & x_M & x_{\text{query}} \\ y_1 & y_2 & \ldots & y_M & 0 \end{pmatrix}. \quad (2)$$

We employ a single-layer linear self-attention (LSA) model $f_{\text{LSA}}(E; \theta)$, defined as:

$$f_{\text{LSA}}(E; \theta) = E + W^{PV} E \frac{E^\top W^{KQ} E}{\rho}, \quad (3)$$

where $\theta = (W^{KQ}, W^{PV})$ denotes the concatenated weight matrices and $\rho$ is a normalization factor. This model has been widely used in previous theoretical analyses of ICL (Von Oswald et al., 2023; Li et al., 2023; Ahn et al., 2023; Zhang et al., 2024; Fu et al., 2025; Fu & Wang, 2026).

The in-context prediction for a query $x_{\text{query}}$ is obtained from the bottom-right entry of the model output. Formally, we define the in-context predictor $f_\mathcal{C} : \mathbb{R}^d \to \mathbb{R}$ as:

$$f_\mathcal{C}(x_{\text{query}}) = [f_{\text{LSA}}(E; \theta)]_{d+1, M+1}. \quad (4)$$

The model is trained via gradient flow $\frac{d}{dt}\theta = -\nabla\mathcal{L}_{pre}(\theta)$ on the population loss over training prompts of length $N$, where the the population loss is:

$$\mathcal{L}_{pre}(\theta) = \frac{1}{2}\mathbb{E}_{w_\tau, x_{\tau, 1:N}, x_{\tau, \text{query}}} \left[f_\mathcal{C}(x_{\tau, \text{query}}) - \langle w_\tau, x_{\tau, \text{query}}\rangle\right]^2 \quad (5)$$

**Objective Function.** To find a recourse $x'$ for the in-context predictor $f_C$, we follow the differentiable framework proposed by Wachter et al. (2017). Given an instance $x$, the objective is to find a recourse $x'$ that minimizes the following joint loss function:

$$\mathcal{L}(x, x') = (f_C(x'))^2 + \lambda \|x - x'\|_2^2, \qquad (6)$$

where $f_C(x')$ is the prediction score produced by the in-context predictor defined in Eq. (4). The second term, $\|x - x'\|_2^2$, represents the squared $L_2$ distance between the original instance and the candidate recourse, serving as a measure of proximity. The hyperparameter $\lambda$ controls the trade-off between achieving the desired outcome (where $f_C(x')$ approaches zero) and ensuring that the recourse remains close to the original input. The optimal recourse is then obtained as $x^* = \arg\min_{x'} \mathcal{L}(x, x')$.

## 4.2. Feasibility Guarantees

We first provide sufficient conditions for the in-context predictor $f_C$ defined in Eq. (4) to provide a universal recourse guarantee. To this end, we distinguish between actionable features, which individuals can modify (e.g., income), and immutable features, which are inherently fixed (e.g., age).

**Proposition 4.1.** *If all features are unbounded, then a predictor induced by in-context learning with at least one actionable feature provides recourse to all individuals.*

**Proposition 4.2.** *If all features are bounded, then a predictor induced by in-context learning with at least one immutable feature may deny recourse to some individuals.*

*Remark* 4.3. Proposition 4.1 and 4.2 highlight how recourse feasibility depends on feature bounds. A meaningful feasibility claim requires a balanced setting of these limits: they must avoid being too loose, which allows for unrealistic changes in actionable variables(e.g., income), while also remaining broad enough to avoid prematurely concluding that recourse is impossible.

## 4.3. Upper bound of Optimal Recourse Cost

To quantify the cost of recourse, we analyze the upper bound of the optimal recourse effort. Given a set of test demonstrations $\{(x_i, y_i)_{i=1}^M\}$, we first derive a closed-form solution for the recourse effort in the in-context learning setting.

**Lemma 4.4** (Optimal Recourse for In-Context Predictors). *Consider the in-context predictor $f_C$ defined in Eq. (4) and the objective function $\mathcal{L}(x, x')$ defined in Eq. (6). For a given instance $x$, the optimal recourse $x^* = x + \delta_{ICL}^*$ that minimizes $\mathcal{L}(x, x')$ is achieved by the perturbation:*

$$\delta_{ICL}^* = -f_C(x) \cdot \left( \Gamma^{-1} S w w^\top S \Gamma^{-1} + \lambda I \right)^{-1} \Gamma^{-1} S w, \quad (7)$$

*where $S := \frac{1}{M} \sum_{i=1}^M x_i x_i^\top$ is the empirical covariance of the prompt covariates, $\Gamma := \left(1 + \frac{1}{N}\right) \Lambda + \frac{1}{N} tr(\Lambda) I_d$ is*

*the pre-training effective covariance matrix, and $f_C(x) = x^\top \Gamma^{-1} S w$ is the continuous prediction score of the in-context learner.*

*Remark* 4.5. When the training context length $N$ and test prompt length $M$ are sufficiently large, we have $\Gamma^{-1} \approx \Lambda^{-1}$ and $S \approx \Lambda$, such that the optimal recourse effort $\delta_{ICL}^*$ converges to the linear solution $\delta_{Linear}^* = -(x^\top w)(w w^\top + \lambda I)^{-1} w$. This establishes that for sufficiently long contexts, ICL-based recourse effectively recovers the optimal strategy of the underlying linear predictor.

In the absence of a fixed demonstration set $\{(x_i, y_i)_{i=1}^M\}$, when the samples are drawn i.i.d. from the underlying distribution, the squared norm of the optimal recourse effort admits the following high-probability upper bound.

**Theorem 4.6** (High-Probability Upper Bound). *Let $\delta \in (0, 1)$. For a given task weight $w$ and covariance $\Lambda$, let the pre-training context length $N$ and test prompt length $M$ be sufficiently large. Then, with probability at least $1 - \delta$ over the sampling of the test prompt, the optimal recourse perturbation $\delta_{ICL}^*$ satisfies:*

$$\|\delta_{ICL}^*\|^2 \leq 2 \ln(4/\delta) \cdot \left[ \mathcal{A} + \frac{\mathcal{B}}{N} + \mathcal{C} \sqrt{\frac{\ln(2d/\delta)}{M}} \right. \\ \left. + \mathcal{O}(N^{-2} + M^{-1}) \right], \qquad (8)$$

*where the geometric coefficients are defined as:*

$$\mathcal{A} = \frac{w^\top \Lambda w}{\|w\|^2}, \mathcal{B} = \frac{2 tr(\Lambda)}{\|w\|^4} \left[ (w^\top \Lambda w)(w^\top \Lambda^{-1} w) - \|w\|^4 \right],$$

*and $\mathcal{C} > 0$ is a constant depending on $\|\Lambda\|_{op}$, $\|\Lambda w\|$ and $\|\Gamma^{-1}\|_{op}$, but independent of $N, M$.*

*Remark* 4.7. Theorem 4.6 highlights three key aspects of the upper bound of the optimal recourse effort: (1) The leading term $2 \ln(\frac{4}{\delta}) \cdot \mathcal{A}$ serves as the dominant contribution to the bound, and can be interpreted as an upper bound on the linear score $f = w^\top x + b$. As $N, M \to \infty$, the additional correction terms vanish and the bound asymptotically tightens to the linear case. (2) The refinement terms $\frac{1}{N} \mathcal{B}$ and $\mathcal{C} \sqrt{\frac{\ln(2d/\delta)}{M}}$ exhibit different convergence rates with respect to $N$ and $M$, reflecting the distinct roles of the number of training demonstrations and the number of test demonstrations. (3) Notably, even as $M \to \infty$, the bound does not converge to the exact score; the remaining deviation is governed by the $\mathcal{O}(\frac{1}{N})$ term, emphasizing that the sample complexity fundamentally depends on $N$, the number of training demonstrations.

## 4.4. Discussions

Theorem 4.6 yields critical practical insights for implementing algorithmic recourse within the In-Context Learning paradigm. In particular, Eq. (8) suggests two theory-supported trends in the analyzed linear-ICL setting. First,

increasing the number of test-time demonstrations $M$ tightens the recourse perturbation bound, while the pre-training context length $N$ is typically fixed and beyond the user's control in frozen black-box models. Second, the perturbation bound depends on the feature dimensionality $d$ through the $\ln(2d/\delta)$ term, suggesting that high-dimensional tabular spaces may require more context examples to maintain stable and low-cost recourse. In real-world deployments involving frozen, black-box large language models, the pre-training context length $N$ is typically immutable and beyond the user's control. Consequently, achieving a lower recourse cost necessitates increasing the number of test-time demonstrations $M$ to refine the decision boundary. However, Eq. (8) reveals a fundamental challenge: the upper bound on recourse perturbation depends on the feature dimensionality $d$ through the $\ln(2d/\delta)$ term. This suggests that as the feature space expands, a larger $M$ is required to maintain the stability and optimality of the recourse solution. To mitigate this effect under limited query budgets, we propose restricting the recourse search to an informative $k$-dimensional subspace ($k \ll d$), thereby artificially reducing the effective dimensionality of the problem. Motivated by this theoretical intuition, we introduce our novel algorithmic framework in the following section.

# 5. Method

Section 4 establishes that algorithmic recourse under ICL is theoretically well-defined and converges to classical solutions as the context size grows. However, these guarantees do not directly yield practical recourse algorithms for real-world models, where predictions are produced by black-box models, gradients are unavailable, and recourse must satisfy strict actionability constraints. In this section, we propose **Adaptive Subspace Recourse for In-Context Learning (ASR-ICL)**, a zeroth-order optimization framework designed to generate sparse and actionable recourse under black-box in-context decision rules. ASR-ICL progressively focuses optimization on a small set of influential features, enabling scalable recourse generation in high-dimensional tabular domains.

## 5.1. Challenges & Motivations

Algorithmic recourse is typically formulated as Eq. (1), which seeks a feasible modification $x' \in \Omega(x)$ that flips the prediction with minimal cost. Most existing approaches assume a fixed trained predictor $f$ and rely on gradient access or smooth optimization. In contrast, under ICL, the decision rule is the context-dependent predictor $f_{\mathcal{C}}$, which is only accessible through expensive black-box queries. As a result, generating recourse becomes a constrained black-box optimization problem in high-dimensional tabular spaces, further complicated by feasibility constraints such as im-

mutability and monotonicity. These challenges motivate a query-efficient framework tailored to ICL-based predictors.

## 5.2. Adaptive Subspace Recourse for In-Context Learning (ASR-ICL)

**Recourse Optimization under Black-box ICL**  To make Eq. (1) computationally tractable for black-box in-context models, we adopt a standard surrogate objective following Wachter et al. (2017). While our theoretical analysis in Section 4 focuses on a squared $\ell_2$ proximity term for analytical convenience, in practice we allow a general recourse cost that captures mixed tabular feature changes. Specifically, we optimize the loss

$$L_{\mathrm{pr}}(x, x') := \big(1 - \mathbb{I}[\hat{y}(x') = y^+]\big) + \lambda c(x, x') \quad (9)$$

where $c(x, x')$ is the recourse cost defined in Section 3, and $\lambda > 0$ balances prediction changes against proximity. This objective can be evaluated solely through queries to $f_{\mathcal{C}}$, without requiring gradients or model internals.

In our black-box setting, we only observe whether the desired target outcome is achieved, represented by the indicator $\mathbb{I}[\hat{y}(x') = y^+]$, which reduces recourse success to a binary signal regardless of whether the underlying task is binary or multi-class. Although this yields a non-smooth objective, it is unavoidable in the strict black-box setting and can be handled via zeroth-order optimization.

**Sparse Subspace Formulation for Recourse**  Directly minimizing Eq. (9) over all $d$ features is inefficient in tabular domains. Moreover, recourse explanations are typically sparse: only a few actionable attributes need to be changed. We therefore restrict optimization to feature subsets. Let $S \subseteq \{1, \ldots, d\}$ with $|S| = k \ll d$, and enforce $x'_{\bar{S}} = x_{\bar{S}}$ to keep the remaining features fixed. The recourse objective in Eq. (9) can be rewritten as follows

$$\min_{S:|S|=k} \quad \min_{x'_S} L\big(x, (x'_S, x_{\bar{S}})\big), \quad (10)$$

which decomposes recourse generation into subspace selection and low-dimensional black-box optimization.

**Adaptive Subspace Selection**  The Eq. (10) requires selecting a feature subset $S$ of size $k$, which is a combinatorial problem in high dimensions. We therefore design an adaptive strategy to efficiently construct informative subspaces during optimization. Each feature $j$ is assigned an importance score $I_j$, which defines the probability of being selected into a subspace:

$$p_{\mathrm{sel}}(j) \propto \exp(I_j). \quad (11)$$

We initialize all scores uniformly, i.e., $I_j = 0$ for all $j$, corresponding to an initial uniform sampling distribution.

At each iteration, we sample a temporary subspace $S_t$ of size $k$ according to $p_{\text{sel}}$:

$$S_t \sim p_{\text{sel}}. \tag{12}$$

After optimizing recourse within the sampled subspace $S_t$, we obtain its achievable objective value:

$$r_t = -\min_{x': x'_{\bar{S}_t} = x_{\bar{S}_t}} L(x, x'), \tag{13}$$

which reflects how effective the features in $S_t$ are for improving recourse. We use this value to update the scores of the involved features via an exponential moving average,

$$I_j \leftarrow (1 - \alpha) I_j + \alpha \frac{r_t}{|S_t|}, \quad \forall j \in S_t. \tag{14}$$

where $\alpha$ is a standard smoothing factor fixed to $\alpha = 0.5$ in all experiments. As a result, the algorithm gradually focuses on a small set of influential features while still exploring alternative choices.

**Optimization Procedure** Since gradients are unavailable for the black-box predictor $f_{\mathcal{C}}$, at each iteration, we sample a feature subspace $S_t \sim p_{\text{sel}}$ and approximately solve the corresponding inner problem in Eq. (10) to obtain the utility $r_t$. We adopt a standard zeroth-order optimizer (Wang et al., 2025b;a), i.e., a black-box optimization method that relies only on function evaluations, to solve each low-dimensional subspace problem. Since our contribution lies in learning informative subspaces rather than in the optimizer itself, the framework is agnostic to the specific solver choice. In practice, we instantiate it with RACOS (Liu et al., 2017; Yu et al., 2016), which is well suited for classification tasks with mixed continuous and categorical features (see more details in Appendix B).

Tabular domains often involve mixed feature types: numerical features take values in bounded continuous intervals, while categorical features belong to finite discrete sets. RACOS naturally supports such mixed search spaces by optimizing over intervals and grids within each subspace. Actionability constraints are enforced by projecting each candidate solution onto the feasible set, $\mathbf{x}' \leftarrow \Pi_\Omega(\mathbf{x}')$, which guarantees that all evaluated recourse instances satisfy immutability and monotonicity requirements. The overall pseudocode of the ASR-ICL is provided in Algorithm 1 in Appendix.

# 6. Experiments

## 6.1. Experimental Setup

**Datasets.** We evaluate ASR-ICL on five real-world tabular datasets covering both binary and multi-class classification tasks. For binary classification, we use Australian

Credit (Quinlan, 1987), COMPAS (Angwin et al., 2022), and Diabetes (Akturk, 2020). For multi-class evaluation, we consider Corporate Rating (Gewerc, 2019) and Student Performance (Kharoua, 2022). These datasets cover diverse high-stakes decision domains and have been widely used in the existing literature on algorithmic recourse (Kanamori et al., 2026; Turbal et al., 2025; Upadhyay et al., 2021).

**Models.** We evaluate ASR-ICL across different models. We consider two specialized tabular ICL models, TabPFN-2.5 (Grinsztajn et al., 2025) and TabICL (Qu et al., 2025), which represent state-of-the-art foundation models designed for tabular prediction. We further include four general large language models, Qwen3-4B-Instruct, Qwen3-8B (Yang et al., 2025), LLaMA-3.2-3B, and LLaMA-3.1-8B (Grattafiori et al., 2024). Finally, we evaluate the closed-source model GPT-4o (Hurst et al., 2024).

**Baseline.** We compare ASR-ICL to the following classical baselines for algorithmic recourse in tabular domains: (i) DiCE (Mothilal et al., 2020), an optimization-based method for generating diverse counterfactual explanations; (ii) Actionable Recourse (AR) (Ustun et al., 2019), a constrained optimization approach for linear decision models; and (iii) FACE (Poyiadzi et al., 2020), a graph-based recourse along the data manifold. We implement all baselines using their original formulations on trained classifiers.

**Metrics.** We consider three metrics in our evaluation: *Validity* measures the fraction of instances for which the generated recourse achieves the desired prediction. *Avg Cost* measures the magnitude of feature changes, computed as the $\ell_1$ distance over normalized continuous features plus a unit cost for each changed categorical feature. *Feature Concentration* captures how strongly the search focuses on a small subset of features, measured by $\exp(H(p))$, where $H(p)$ is the entropy of the learned sampling distribution. See more details in Appendix D.3.

**Implementation Details.** All experiments are conducted under the same query budget and feasibility constraints. For ASR-ICL, we use a fixed subspace size $k$ and a RACOS-based zeroth-order solver for inner optimization. Unless otherwise stated, all hyperparameters follow the default settings described in Appendix D.4.

## 6.2. Results on ASR-ICL

Table 1 reports the performance of ASR-ICL across datasets, compared with classical methods on trained models. While DiCE and FACE achieve perfect validity on trained models, they often require substantially larger recourse costs (e.g., cost 8.96 on Australian Credit), indicating dense feature changes that may be less actionable in practice. AR produces cheaper recourse but at the expense of reduced

*Table 1.* Performance comparison between ASR-ICL and existing methods. All ICL-based results are reported under 32-shot context. We report validity and average cost with mean $\pm$ standard deviation.

| Model | Method | Australian Credit | | COMPAS | | Diabetes | |
|---|---|---|---|---|---|---|---|
| | | Validity | Avg Cost | Validity | Avg Cost | Validity | Avg Cost |
| **Trained Models** | | | | | | | |
| MLP | DiCE | $1.00 \pm 0.00$ | $8.96 \pm 1.14$ | $1.00 \pm 0.00$ | $7.51 \pm 0.76$ | $1.00 \pm 0.00$ | $5.02 \pm 0.25$ |
| Linear Model | AR | $0.82 \pm 0.08$ | $1.76 \pm 0.16$ | $0.94 \pm 0.18$ | $3.44 \pm 0.28$ | $0.72 \pm 0.07$ | $1.52 \pm 0.19$ |
| Linear Model | FACE | $1.00 \pm 0.00$ | $6.18 \pm 0.27$ | $1.00 \pm 0.00$ | $6.51 \pm 0.29$ | $1.00 \pm 0.00$ | $5.13 \pm 0.15$ |
| **ICL-based Models** | | | | | | | |
| TabPFN-2.5 | | $1.00 \pm 0.00$ | $3.83 \pm 0.04$ | $1.00 \pm 0.00$ | $2.76 \pm 0.05$ | $1.00 \pm 0.00$ | $2.78 \pm 0.08$ |
| TabICL | | $1.00 \pm 0.00$ | $4.47 \pm 0.15$ | $0.81 \pm 0.05$ | $3.44 \pm 0.43$ | $1.00 \pm 0.00$ | $2.80 \pm 0.16$ |
| Qwen3-4B-Instruct | | $0.79 \pm 0.15$ | $3.01 \pm 0.06$ | $1.00 \pm 0.01$ | $2.67 \pm 0.06$ | $0.97 \pm 0.03$ | $1.58 \pm 0.12$ |
| Qwen3-8B | ASR-ICL | $0.87 \pm 0.08$ | $2.94 \pm 0.15$ | $0.99 \pm 0.01$ | $2.55 \pm 0.09$ | $0.84 \pm 0.04$ | $1.50 \pm 0.09$ |
| LLaMA-3.2-3B | | $1.00 \pm 0.00$ | $2.99 \pm 0.07$ | $1.00 \pm 0.00$ | $2.43 \pm 0.08$ | $0.98 \pm 0.03$ | $1.67 \pm 0.09$ |
| LLaMA-3.1-8B | | $0.80 \pm 0.08$ | $2.96 \pm 0.05$ | $0.97 \pm 0.04$ | $2.40 \pm 0.07$ | $1.00 \pm 0.00$ | $1.50 \pm 0.04$ |
| GPT-4o | | $0.99 \pm 0.03$ | $4.75 \pm 0.09$ | $0.78 \pm 0.07$ | $3.62 \pm 0.12$ | $0.71 \pm 0.05$ | $4.31 \pm 0.09$ |

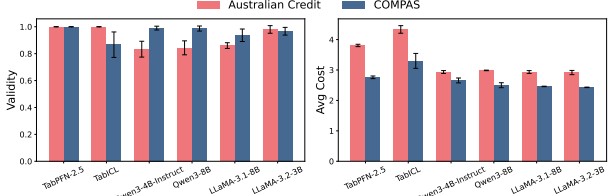

*Figure 1.* Effect of the number of context instances on recourse quality under ICL. Increasing the number of instances improves stability and reduces recourse cost.

validity. In contrast, ASR-ICL achieves consistently strong validity, including both tabular ICL models and general LLMs, while maintaining a low recourse cost. For example, on Australian Credit, ASR-ICL attains perfect validity with TabPFN at an average cost of 3.83, significantly lower than DiCE and FACE. Similar trends hold for COMPAS and Diabetes, where ASR-ICL produces valid recourse with costs around $2 \sim 3$, despite operating in a fully black-box, context-conditioned setting. Performance remains robust for most open-source LLMs (e.g., LLaMA and Qwen), though GPT-4o exhibits reduced validity on some datasets, highlighting the additional challenge of recourse under weaker or noisier in-context decision rules. Overall, these results demonstrate that ASR-ICL enables actionable algorithmic recourse for modern models, bridging the gap between classical recourse methods designed for trained models and emerging in-context learning models.

### 6.3. Empirical Validation of Theoretical Analysis

We empirically validate the theoretical predictions in Section 4 by examining how recourse quality evolves as the

*Figure 2.* Stability of ASR-ICL under context resampling. We repeat experiments with independently sampled 32-shot context instances while fixing test instances. Bars show mean validity and average recourse cost, with error bars indicating standard deviation.

number of in-context examples increases. Our analysis is motivated by Lemma 4.4 and Theorem 4.6, which suggest that as the context size grows, model behave increasingly like classical linear models, leading to more stable and lower-cost recourse. Figure 1 summarizes validity and average recourse cost as a function of the number of in-context instances across different models. When only a few context examples are provided (e.g., 4 or 8 shots), recourse

*Table 2.* Recourse performance on multi-class tabular tasks. Corporate Rating uses a 24-shot context and Student Performance uses a 40-shot context. We report validity and average cost with mean ± standard deviation.

| Model | Corporate Rating | | Student Performance | |
|---|---|---|---|---|
| | Validity | Avg Cost | Validity | Avg Cost |
| TabPFN-2.5 | $0.98 \pm 0.02$ | $4.79 \pm 0.19$ | $1.00 \pm 0.00$ | $3.63 \pm 0.03$ |
| TabICL | $0.99 \pm 0.02$ | $5.79 \pm 0.25$ | $1.00 \pm 0.00$ | $3.42 \pm 0.07$ |
| Qwen3-4B-Instruct | $0.94 \pm 0.04$ | $3.54 \pm 0.06$ | $0.55 \pm 0.10$ | $3.37 \pm 0.06$ |
| Qwen3-8B | $0.73 \pm 0.04$ | $3.55 \pm 0.09$ | $0.42 \pm 0.22$ | $3.18 \pm 0.10$ |
| LLaMA-3.2-3B | $0.95 \pm 0.07$ | $3.01 \pm 0.10$ | $0.45 \pm 0.06$ | $3.18 \pm 0.03$ |
| LLaMA-3.1-8B | $0.50 \pm 0.24$ | $3.15 \pm 0.11$ | $0.43 \pm 0.08$ | $3.24 \pm 0.02$ |
| GPT-4o | $0.72 \pm 0.07$ | $4.87 \pm 0.18$ | $0.84 \pm 0.05$ | $4.59 \pm 0.08$ |

validity is noticeably less stable and varies substantially across models. As the context size increases, validity consistently improves and becomes more concentrated, eventually approaching the linear-model reference performance. A similar trend is observed for recourse cost. Across all models, the average cost decreases as more context instances are available, indicating that fewer or smaller feature modifications are needed to reach the desired outcome. Recourse costs gradually become comparable to the linear-model reference, and in some cases even slightly lower. This can occur because modern ICL models may admit sparser feature modifications under the same cost metric, leading to more efficient actionable recourse than fixed decision rule.

Finally, Figure 2 demonstrates that ASR-ICL produces highly consistent recourse solutions across independently resampled context sets with randomized instance selection and ordering. Both validity and recourse cost exhibit only small variance, suggesting that the proposed method is robust to fluctuations in context composition and does not rely on a specific demonstration choice.

### 6.4. Multi-class Recourse

While most existing recourse methods focus on binary classification, many real-world decision systems are inherently multi-class, such as credit rating assignment and educational outcome prediction. ASR-ICL naturally extends to multi-class tabular settings by optimizing toward a desired target class without requiring task-specific reductions. Table 2 reports multi-class recourse performance. Across both tasks, ASR-ICL achieves near-perfect validity for specialized tabular ICL predictors such as TabPFN and TabICL, with moderate recourse costs around $3 \sim 6$. These results indicate that the proposed framework remains effective beyond the binary setting. For general LLMs, recourse validity is lower, particularly on Student Performance, reflecting the increased difficulty of inducing reliable multi-class decision boundaries from limited tabular context. Nevertheless, ASR-ICL continues to produce feasible low-cost recourse solutions when successful, demonstrating robustness of the framework across diverse black-box models.

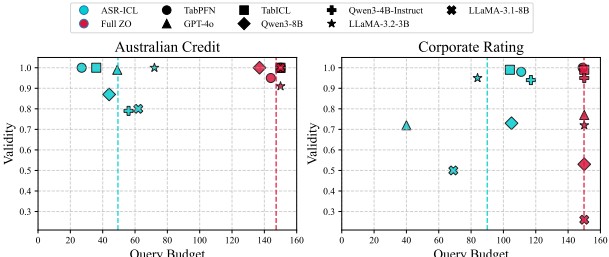

*Figure 3.* Validity versus query budget across different tasks (Australian Credit and Corporate Rating). Each point corresponds to a model–method pair, and vertical dashed lines indicate the mean query budget required by each method. ASR-ICL consistently achieves comparable validity with substantially fewer model queries. See Appendix F.1 for additional results.

### 6.5. Ablation and Sensitive Analysis

We provide additional analysis of ASR-ICL regarding efficiency, feature behavior, and robustness to key hyperparameters. Figure 3 shows that across two tabular decision tasks (Australian Credit and Corporate Rating), ASR-ICL achieves high recourse validity with substantially fewer black-box model queries than a non-adaptive full-space zeroth-order baseline (Full ZO), reducing the average query budget by roughly $2$–$3\times$ across different predictors. In addition, ASR-ICL achieves not only query savings but also substantially lower recourse costs than Full ZO, as detailed in Appendix F.1. Moreover, ASR-ICL progressively concentrates its search on a small subset of important features, promoting sparse and efficient recourse discovery. Feature concentration dynamics are reported in Appendix F.2. Sensitivity studies in Appendix F.3 indicate that ASR-ICL remains stable under moderate variations of the trade-off parameter $\lambda$ and the subspace size $k$.

### 6.6. Case Study

We present a qualitative case study on a real instance from the Diabetes dataset, where the original prediction is unfavorable and the goal is to obtain a feasible recourse. Table 7 compares recourse modifications produced by classical baselines (DiCE, AR, FACE), Full ZO, and ASR-ICL. DiCE

modifies immutable attributes and is thus infeasible, while AR, FACE, and Full ZO typically require changing multiple features. In contrast, ASR-ICL flips the prediction by modifying only two actionable attributes (Glucose and BMI), yielding a sparser and more interpretable recourse suggestion. This example illustrates how ASR-ICL promotes recourse explanations that align with human expectations and practical actionability.

# 7. Conclusion

We presented the first study of algorithmic recourse of in-context learning on tabular data. Our results establish theoretical feasibility and convergence guarantees. We propose ASR-ICL, a query-efficient zeroth-order framework for generating sparse and actionable recourse under black-box ICL models. Experiments on diverse real-world datasets validate the effectiveness of ASR-ICL in both binary and multi-class settings.

# Acknowledgements

This work is partially supported by the MBZUAI Research Fund BF0100. Di Wang and Shaopeng Fu are supported in part by the funding BAS/1/1689-01-01,RGC/3/7125-01-01, FCC/1/5940-20-05, FCC/1/5940-06-02, and King Abdullah University of Science and Technology (KAUST) – Center of Excellence for Generative AI, under award number 5940 and a gift from Google.

# Impact Statement

This paper presents work whose goal is to advance the field of Machine Learning. There are many potential societal consequences of our work, none which we feel must be specifically highlighted here.

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

# A. Omit Proof

## A.1. Proof of Proposition 4.1 & 4.2

*Proof of 4.1.* Consider the predictor induced by in-context learning with closed-form solution by Zhang et al. (2024)

$$f_{\text{ICL}}(x) = \text{sign}\left(x_{\text{query}}^\top \Gamma^{-1}\left(\frac{1}{M}\sum_{i=1}^{M} x_i x_i^\top\right) w\right).$$

Let $j$ denote the index of a feature that can be increased or decreased arbitrarily. Assume, without loss of generality, that $\left[\Gamma^{-1}\left(\frac{1}{M}\sum_{i=1}^{M} x_i x_i^\top\right) w\right]_j > 0$.

For any feature vector $x$ such that

$$f_{\text{ICL}}(x) = \text{sign}\left(x^\top \Gamma^{-1}\left(\frac{1}{M}\sum_{i=1}^{M} x_i x_i^\top\right) w\right) = -1,$$

the set of feasible actions from $x$ must contain an action vector $a = [0, a_1, a_2, \ldots, a_d]$ such that $a_j >$
$-\dfrac{x^\top \Gamma^{-1}\left(\frac{1}{M}\sum_{i=1}^{M} x_i x_i^\top\right) w}{\left[\Gamma^{-1}\left(\frac{1}{M}\sum_{i=1}^{M} x_i x_i^\top\right) w\right]_j}$, $\qquad a_k = 0$ for all $k \neq j$.

Thus, in-context learning provides $x$ with recourse since $(x + a)^\top \Gamma^{-1}\left(\frac{1}{M}\sum_{i=1}^{M} x_i x_i^\top\right) w > 0$, and $f_{\text{ICL}}(x + a) = \text{sign}\left((x + a)^\top \Gamma^{-1}\left(\frac{1}{M}\sum_{i=1}^{M} x_i x_i^\top\right) w\right) = +1$.

Since our choice of $x$ was arbitrary, the result holds for all $x \in H^-$. Hence, the predictor induced by in-context learning provides recourse to all individuals in the target population. $\qquad\square$

*Proof of Proposition 2.* Suppose there are $d$ actionable binary features $x_j \in \{0, 1\}$ for $j \in \{1, \ldots, d\}$ and one immutable binary feature $x_{d+1} \in \{0, 1\}$. Assume the in-context predictor has the closed form

$$f_{\text{ICL}}(x) = \text{sign}\left(x_{\text{query}}^\top \Gamma^{-1}\left(\frac{1}{M}\sum_{i=1}^{M} x_i x_i^\top\right) w\right),$$

and that the corresponding coefficients satisfy

$$\left[\Gamma^{-1}\left(\frac{1}{M}\sum_{i=1}^{M} x_i x_i^\top\right) w\right]_j = 1 \quad \text{for } j = 1, \ldots, d, \qquad \left[\Gamma^{-1}\left(\frac{1}{M}\sum_{i=1}^{M} x_i x_i^\top\right) w\right]_{d+1} = \alpha,$$

with $\alpha < -1$. (Such coefficients can be realized by an appropriate choice of $w$ and demonstration set $\{x_i\}_{i=1}^{M}$.) Then the ICL score equals

$$x^\top \Gamma^{-1}\left(\frac{1}{M}\sum_{i=1}^{M} x_i x_i^\top\right) w = \sum_{j=1}^{d} x_j + \alpha x_{d+1}.$$

If the decision threshold is $d$ (as in the linear example), then for any $x$ with $x_{d+1} = 1$ we have

$$\sum_{j=1}^{d} x_j + \alpha x_{d+1} = \sum_{j=1}^{d} x_j + \alpha < \sum_{j=1}^{d} x_j - 1 \leq d - 1,$$

so such $x$ cannot reach the threshold and therefore has no recourse.

Thus, even though the predictor is realized by in-context learning, the presence of a sufficiently negative coefficient on an immutable feature can cause the predictor to deny recourse to some individuals, exactly as in the linear classifier case.

$\qquad\square$

## A.2. Proof of Lemma 4.4

**Lemma A.1** (Optimal Recourse for Linear classifier). *Given a scoring function with weights $\mathbf{w}$, minimizing the objective in (6) yields the optimal recourse $\mathbf{x}^* = \mathbf{x} + \delta^*_{linear}$ for an input $\mathbf{x}$ such that:*

$$\delta^*_{linear} = -f(\mathbf{x} \cdot (\mathbf{w}\mathbf{w}^T + \lambda I)^{-1}\mathbf{w},$$

*where $f(x) = \mathbf{w}^T\mathbf{x} + b$ is a local linear score approximation, and $\lambda$ is a given hyperparameter.*

*Proof of Lemma 4.4.* Recall that Lemma A.1 gives the optimal recourse for a linear scoring function $f(x) = w^\top x + b$ as

$$\delta^*_{linear} = -f(x) \cdot (ww^\top + \lambda I)^{-1}w,$$

In the in-context learning setting, the query prediction admits the closed-form solution by Zhang et al. (2024):

$$\hat{y}_{\text{query}} = x_{\text{query}}^\top \Gamma^{-1}\Big(\frac{1}{M}\sum_{i=1}^{M} y_i x_i\Big)w,$$

Comparing with the linear score $f(x) = w^\top x$, we can identify an effective weight vector for the ICL predictor:

$$w_{\text{ICL}} = \Gamma^{-1}\Big(\frac{1}{M}\sum_{i=1}^{M} y_i x_i\Big),$$

so that $\hat{y}_{\text{query}} = x_{\text{query}}^\top w_{\text{ICL}}$.

Applying Lemma A.1 with $w_{\text{ICL}}$ in place of $w$, the optimal recourse becomes

$$\delta^*_{\text{ICL}} = (s - \hat{y}_{\text{query}}) \cdot \big(w_{\text{ICL}}w_{\text{ICL}}^\top + \lambda I\big)^{-1}w_{\text{ICL}}w.$$

Substituting the expression for $w_{\text{ICL}}$ explicitly in terms of the training prompts and $\Gamma$, we obtain

$$\delta^*_{\text{ICL}} = m \cdot \Big(\Gamma^{-1}\big(\tfrac{1}{M}\sum_{i=1}^{M} y_i x_i\big)ww^\top\big(\tfrac{1}{M}\sum_{i=1}^{M} y_i x_i\big)^\top\Gamma^{-1} + \lambda I\Big)^{-1}\Gamma^{-1}\big(\tfrac{1}{M}\sum_{i=1}^{M} y_i x_i\big)w.$$

$\square$

## A.3. Proof of Theorem 4.6

**Lemma A.2** (Gaussian Tail Bound). *Let $Z \sim \mathcal{N}(0, \sigma^2)$. Then for any $\delta \in (0, 1)$,*

$$\Pr(Z^2 \geq 2\sigma^2 \ln(2/\delta)) \leq \delta.$$

**Lemma A.3** (Matrix Bernstein Concentration (Tropp et al., 2015)). *Let $X_i$ be independent, mean-zero, symmetric random matrices with $\|X_i\|_{\text{op}} \leq R$. Define $S = \sum_i X_i$. Then, for any $t > 0$,*

$$\Pr\big(\|S\|_{\text{op}} \geq t\big) \leq 2d\exp\Big(\frac{-t^2/2}{\sigma^2 + Rt/3}\Big),$$

*where $\sigma^2 = \|\sum_i \mathbb{E}[X_i^2]\|_{\text{op}}$.*

*Proof of Theorem 4.6.* We define $S := \frac{1}{M}\sum_{i=1}^{M} x_i x_i^\top \in \mathbb{R}^{d \times d}$.. The expression in lemma 4.4 can further be simplified:

$$\delta^* = \frac{m}{\lambda}\left(I - \frac{\Gamma^{-1}S\mathbf{w}\mathbf{w}^T S\Gamma^{-1}}{\lambda + \|\Gamma^{-1}S\mathbf{w}\|_2^2}\right)\mathbf{w}$$

$$= \frac{m}{\lambda}\left(I\Gamma^{-1}S\mathbf{w} - \Gamma^{-1}S\mathbf{w}\frac{\|\Gamma^{-1}S\mathbf{w}\|_2^2}{\lambda + \|\Gamma^{-1}S\mathbf{w}\|_2^2}\right)$$

$$= \frac{m}{\lambda}\cdot\frac{\lambda}{\lambda + \|\Gamma^{-1}S\mathbf{w}\|_2^2}\cdot\Gamma^{-1}S\mathbf{w} = \frac{m}{\lambda + \|\Gamma^{-1}S\mathbf{w}\|_2^2}\cdot\Gamma^{-1}S\mathbf{w},$$

where $m := s - f(\mathbf{x})$. The first equation comes form Sherman-Morrison Formula. Finally, we note that as $\lambda \to 0$, we have:

$$\delta^{**} = \frac{m}{\|\Gamma^{-1}S\mathbf{w}\|_2^2}\cdot\Gamma^{-1}S\mathbf{w}.$$

Thus, by define $u := \frac{\Gamma^{-1}S\mathbf{w}}{\|\Gamma^{-1}S\mathbf{w}\|}$, we can simplify $\|\delta_{ICL}\|^2$ as:

$$\|\delta_{ICL}\|^2 = \delta_{ICL}\cdot\delta_{ICL}^\top = \frac{x^\top\Gamma^{-1}S\mathbf{w}\mathbf{w}^\top S\Gamma^{-1}x}{\|\Gamma^{-1}S\mathbf{w}\|_2^2} = (u^\top x)^2.$$

Conditioned on $\Gamma^{-1}S$ (and hence on $S$), $u^\top x \sim \mathcal{N}(0, u^\top\Lambda u)$, by the standard property of Gaussian vectors. Using the tail bound for a one-dimensional Gaussian by applying lemma A.2 to $u^\top x$ conditioned on $\Gamma^{-1}S$, we get

$$\Pr\left((u^\top x)^2 \le 2(u^\top\Lambda u)\ln(2/\delta_2)\,\Big|\,\Gamma^{-1}S\right) \ge 1 - \delta_2.$$

Next, we want to compute the upper bound of $u^\top\Lambda u$. Applying Lemma A.3 to $X_i = x_i x_i^\top - \Lambda$ gives

$$\Pr\left(\|S - \Lambda\|_{\mathrm{op}} \le t\right) \ge 1 - \delta_1, \quad t = C_0\|\Lambda\|_{\mathrm{op}}\frac{(\sqrt{M\ln(2d/\delta_1)} + \ln(2d/\delta_1))}{M} \sim C_0\|\Lambda\|_{\mathrm{op}}\sqrt{\frac{\ln(2d/\delta_1)}{M}}$$

On the event $\|S - \Lambda\|_{\mathrm{op}} \le t$, we expand

$$u^\top\Lambda u = \frac{w^\top\Gamma^{-1}(\Lambda + \Delta)\Lambda(\Lambda + \Delta)\Gamma^{-1}w}{w^\top\Gamma^{-1}(\Lambda + \Delta)^2\Gamma^{-1}w}, \quad \Delta := S - \Lambda.$$

**Bound the correction term from $S$.** In the numerator, the leading term is $\Gamma^{-1}\Lambda^3\Gamma^{-1}$, and the cross term is $\Gamma^{-1}\Lambda^2\Delta\Gamma^{-1}$. The ratio of cross term and leading term is

$$\frac{\|\Gamma^{-1}\Lambda^2\Delta\Gamma^{-1}\|_{op}}{\|\Gamma^{-1}\Lambda^3\Gamma^{-1}\|_{op}} \le O(\frac{\|\Delta\|_{op}}{\|\Lambda\|_{op}}) \sim O(\frac{C_0\|\Lambda\|_{\mathrm{op}}\sqrt{\frac{\ln(2d/\delta_1)}{M}}}{\|\Lambda\|_{op}}) \sim O(C_0\sqrt{\frac{\ln(2d/\delta)}{M}}).$$

Similarly, in the denominator, the leading term is $\Gamma^{-1}\Lambda^2\Gamma^{-1}$, and the cross term is $\Gamma^{-1}\Lambda\Delta\Gamma^{-1}$, and the ratio is $O(\frac{1}{M})$. Therefore, we have

$$u^\top\Lambda u = \frac{w^\top\Gamma^{-1}(\Lambda + \Delta)\Lambda(\Lambda + \Delta)\Gamma^{-1}w}{w^\top\Gamma^{-1}(\Lambda + \Delta)^2\Gamma^{-1}w}$$

$$\le \frac{w^\top\Gamma^{-1}\Lambda^3\Gamma^{-1}w}{w^\top\Gamma^{-1}\Lambda^2\Gamma^{-1}w} + KC_0\sqrt{\frac{\ln(2d/\delta)}{M}}$$

where $K$ is a constant depending on $\|\Lambda\|_{\mathrm{op}}$, $\|\Lambda w\|$ and $\Gamma^{-1}$.

**Bound the correction term from $\Gamma^{-1}$.** When $N$ is very large, the cross term and quadratic term are much smaller than the leading term $N^2$, and can be handled using the Neumann series expansion:

$$(\Gamma^{-1}\Lambda) = \left(\Lambda + \frac{1}{N}(\Lambda + \tau I)\right)^{-1} \approx \Lambda^{-1} - \frac{1}{N}(\Lambda^{-1} + \tau\Lambda^{-2}) + O(1/N^2),$$

where $\tau = \text{tr}(\Lambda)$.

Substituting into the numerator and denominator, we get:

$$\frac{w^\top\Gamma^{-1}\Lambda^3\Gamma^{-1}w}{w^\top\Gamma^{-1}\Lambda^2\Gamma^{-1}w} = \frac{w^\top\Lambda w}{\|w\|^2} + \frac{1}{N}\frac{2\tau}{\|w\|^4}\left((w^\top\Lambda w)(w^\top\Lambda^{-1}w) - \|w\|^4\right) + O(1/N^2),$$

Taking the union bound over the events from lemma A.2 ($x$) and lemma A.3 ($S$) with $\delta_1 = \delta_2 = \delta/2$, we obtain

$$\Pr\left(\|\delta_{ICL}\|^2 \leq 2\ln(4/\delta)\left(\mathcal{A} + \frac{1}{N}\mathcal{B} + \mathcal{C}\sqrt{\frac{\ln(2d/\delta)}{M}} + O(\frac{1}{N^2} + \frac{1}{M}))\right)\right) \geq 1 - \delta,$$

which is exactly equation (8) $\qquad\qquad\qquad\qquad\qquad\qquad\qquad\qquad\qquad\qquad\qquad\qquad$ $\square$

## B. Zeroth-Order Optimization

This section provides additional background on the zeroth-order optimization procedure used as the numerical solver in ASR-ICL.

**Zeroth-Order Optimization.** Recourse generation under in-context learning requires optimizing objectives of the form

$$\min_{x'}\ L(x, x'),$$

where the predictor $f_{\mathcal{C}}$ is induced by a black-box model. In this setting, gradients of $L$ are not available, and each evaluation requires an expensive forward query to the underlying large language model. As a result, standard gradient-based or differentiable recourse methods are not applicable.

Zeroth-order optimization (ZOO) refers to a class of derivative-free optimization methods that rely only on function evaluations rather than gradient information. Such methods are widely used for black-box and non-convex objectives, and are particularly suitable when the model is only accessible through queries. In ASR-ICL, ZOO is used as an inner solver restricted to a low-dimensional feature subspace, significantly reducing query complexity.

**RACOS.** In our implementation, we instantiate the inner zeroth-order solver using RACOS (Randomized Coordinate Shrinking Optimization Strategy) (Yu et al., 2016; Liu et al., 2017). RACOS is a randomized derivative-free optimizer designed for non-convex optimization over mixed continuous and discrete domains. It iteratively samples candidate solutions, evaluates their objective values, and refines the search distribution toward promising regions of the space.

RACOS is particularly well suited for recourse in tabular domains for two reasons. First, tabular inputs often contain both numerical and categorical attributes. Continuous features can be optimized over bounded intervals, while categorical variables naturally correspond to finite discrete grids. RACOS directly supports such mixed search spaces without requiring relaxation or surrogate gradients. Second, the recourse objective under ICL is typically non-smooth due to discrete feature changes and the implicit nature of prompt-conditioned predictors, making randomized black-box optimization a practical choice.

**Feasibility Projection.** To ensure that suggested recourse actions remain actionable, all candidate solutions produced by the solver are projected onto the feasible set $\Omega(x)$ before evaluation. This projection enforces feature-level constraints such as immutability and monotonicity (e.g., only allowing increases in income or preserving fixed attributes). Therefore, every evaluated recourse instances satisfies the feasibility requirements of algorithmic recourse.

## C. Additional Algorithm Details

For completeness, we provide the full pseudocode of ASR-ICL in Algorithm 1.

---

**Algorithm 1** ASR-ICL: Adaptive Subspace Recourse for In-Context Learning

---

**Require:** Instance $\mathbf{x}$, objective $L_{\mathrm{pr}}(\mathbf{x}, \mathbf{x}')$, feasibility projection $\Pi_\Omega(\cdot)$, subspace size $k$, smoothing factor $\alpha$, stopping threshold $\tau$, query budget $B$

**Ensure:** Recourse instance $\mathbf{x}^\star$

1: Initialize importance scores $I_j \leftarrow 0$ for all mutable features $j$
2: Initialize sampling distribution $p_{\mathrm{sel}}(j) = \frac{\exp(I_j)}{\sum_\ell \exp(I_\ell)}$
3: $\mathbf{x}^\star \leftarrow \mathbf{x}, \quad L_{\mathrm{pr}}^\star \leftarrow L_{\mathrm{pr}}(\mathbf{x}, \mathbf{x})$
4: **while** $L_{\mathrm{pr}}^\star > \tau$ **and** $B > 0$ **do**
5:      Sample a temporary subspace $S_t$ of size $k$ according to $p_{\mathrm{sel}}$
6:      Use a zeroth-order optimizer to approximately solve

$$\min_{\mathbf{x}': \mathbf{x}'_{\bar{S}_t} = \mathbf{x}_{\bar{S}_t}} L_{\mathrm{pr}}(\mathbf{x}, \mathbf{x}')$$

     and obtain a candidate solution $\tilde{\mathbf{x}}$
7:      $\tilde{\mathbf{x}} \leftarrow \Pi_\Omega(\tilde{\mathbf{x}})$
8:      $r_t \leftarrow -L_{\mathrm{pr}}(\mathbf{x}, \tilde{\mathbf{x}})$
9:      **if** $L_{\mathrm{pr}}(\mathbf{x}, \tilde{\mathbf{x}}) < L_{\mathrm{pr}}^\star$ **then**
10:          $\mathbf{x}^\star \leftarrow \tilde{\mathbf{x}}$
11:          $L_{\mathrm{pr}}^\star \leftarrow L_{\mathrm{pr}}(\mathbf{x}, \tilde{\mathbf{x}})$
12:      **end if**
13:      **for all** $j \in S_t$ **do**
14:          $I_j \leftarrow (1 - \alpha)I_j + \alpha \cdot \frac{r_t}{|S_t|}$
15:      **end for**
16:      Update $p_{\mathrm{sel}}(j) = \frac{\exp(I_j)}{\sum_\ell \exp(I_\ell)}$
17:      Decrease $B$ according to the number of model evaluations used
18: **end while**
19: **return** $\mathbf{x}^\star$

---

# D. Experimental Setup

## D.1. Datasets

We evaluate algorithmic recourse under ICL on five real-world tabular classification benchmarks, covering both binary and multi-class decision-making settings. These datasets span high-stakes domains where recourse is particularly relevant, such as credit approval, criminal risk assessment, healthcare prediction, and educational and financial evaluation. For all binary datasets, we generate recourse toward the favorable outcome, i.e., the positive label $y^+ = 1$. For the multi-class datasets, recourse is generated toward the most favorable class: rating level 2 for Corporate Rating, and grade A for Student Performance.

**Australian Credit.** The Australian Credit dataset is a widely used benchmark dataset for binary credit decision-making. The objective is to predict whether a credit card application is approved or rejected. Each instance is described by 14 input features, denoted as $A1$–$A14$, corresponding to anonymized personal and financial attributes. Following the original dataset specification, the feature set includes 6 categorical and 8 numerical variables. The prediction target is a binary label indicating the credit approval outcome.

**COMPAS.** The COMPAS dataset is a widely used benchmark dataset for criminal justice risk assessment. It consists of instances where the objective is to predict whether a defendant will reoffend within a two-year period. In this work, we use the `two-years-recidivity-no-race` version, which removes the sensitive attribute race. Each instance is described by 12 features capturing demographic information, criminal history, and COMPAS risk scores. The feature set includes 3 binary categorical indicators and 9 numerical attributes. The prediction target is a binary label indicating whether an individual recidivates within two years.

**Diabetes.** The Diabetes dataset is a widely used benchmark dataset for binary prediction in healthcare decision-making. It consists of 768 instances, where the objective is to predict whether a patient shows signs of diabetes. Each instance is described by 8 numerical features capturing diagnostic measurements, including pregnancies, glucose level, blood pressure, skin thickness, insulin, body mass index (BMI), diabetes pedigree function, and age. The prediction target is a binary label indicating diabetes onset.

**Corporate Rating.** The Corporate Rating dataset is a benchmark dataset for multi-class classification in financial risk assessment. It consists of 2029 corporate credit rating records collected from major rating agencies for publicly traded companies. The objective is to predict a company's credit rating category based on its financial characteristics. Each instance is described by 30 features. Among them, 25 are numerical financial indicators derived from liquidity, profitability, debt, operating performance, and cash-flow ratios. The dataset also includes several categorical attributes such as the company sector and the issuing rating agency. The prediction target is a multi-class label corresponding to standard corporate credit rating levels (e.g., AAA to D). Due to the imbalance of data labels, we grouped AAA, AA, and A into one category, BBB and BB into another, and the rest into the last category.

**Student Performance.** The Student Performance dataset is a benchmark dataset for multi-class classification in educational outcome prediction. It consists of 2392 high school student records, where the objective is to predict academic performance levels based on demographic and behavioral factors. Each instance is described by 13 features capturing academic indicators (e.g., GPA, study time, and absences), as well as background attributes such as age and parental involvement. The prediction target is a multi-class label corresponding to different grade categories (e.g., A–F).

### D.2. Baseline Details

In this section, we provide additional details on the classical algorithmic recourse baselines used in our experiments. These methods are designed for fixed trained models, and represent widely adopted paradigms in the recourse literature.

Given an instance $x \in \mathbb{R}^d$ with an unfavorable prediction $f(x) = 0$, the goal of algorithmic recourse is to compute a recourse instance $x'$ such that $f(x') = 1$, while ensuring that the required changes are minimal and actionable.

#### D.2.1. DICE: DIVERSE COUNTERFACTUAL EXPLANATIONS

DiCE (Mothilal et al., 2020) is an optimization-based and model-agnostic method for generating a set of diverse counterfactual recourse solutions. Given an input instance $x$, DiCE aims to produce $K$ valid counterfactuals $\{x'_1, \ldots, x'_K\}$ that flip the classifier prediction while remaining close to the original instance.

DiCE solves the following objective:

$$\min_{\{x'_k\}_{k=1}^K} \sum_{k=1}^K \left[ \mathcal{L}_{\text{cf}}(x'_k) + \lambda_1 \, d(x, x'_k) \right] - \lambda_2 \det(K), \tag{15}$$

where $\mathcal{L}_{\text{cf}}$ is a hinge-style loss encouraging prediction flipping, $d(\cdot, \cdot)$ is a proximity distance normalized by feature-wise MAD statistics, and $\det(K)$ is a determinantal point process (DPP) term promoting diversity among the generated counterfactuals. DiCE is typically optimized using gradient-based methods such as Adam, and may include an additional sparsity-enhancement post-processing step.

#### D.2.2. ACTIONABLE RECOURSE (AR)

Actionable Recourse (AR) (Ustun et al., 2019) formulates recourse under linear decision models as a constrained optimization problem over feasible actions. Given a linear classifier $f(x) = \text{sign}(w^\top x + b)$, AR computes the minimum-cost actionable intervention that moves an individual across the decision boundary.

The recourse problem is:

$$\min_{a \in A(x)} c(a) \quad \text{s.t.} \quad w^\top(x + a) + b \geq 0, \tag{16}$$

where $a$ denotes feature-level actions restricted to an action set $A(x)$, which encodes immutability, discrete changes, and directional constraints. AR is solved via mixed-integer programming (MIP), yielding globally optimal solutions and allowing

certification when no feasible recourse exists. The method can additionally generate multiple distinct recourse options (*flipsets*) by enumerating alternative minimal-cost actions.

### D.2.3. FACE: FEASIBLE AND ACTIONABLE COUNTERFACTUAL EXPLANATIONS

FACE (Poyiadzi et al., 2020) is a graph-based recourse method that generates counterfactual explanations by searching along the data manifold. Instead of directly optimizing in feature space, FACE constructs a graph $G = (V, E)$ where nodes correspond to training samples and edges connect nearby instances (e.g., via $k$-NN or $\epsilon$-neighborhood graphs).

Given an instance $x$, FACE identifies a sequence of feasible transitions

$$x \to x_1 \to \cdots \to x', \tag{17}$$

such that the endpoint $x'$ attains the desired prediction and intermediate steps remain in high-density regions of the data distribution. Edge weights incorporate both distance and density penalties, encouraging shortest paths that avoid low-density (off-manifold) areas. The final counterfactual is obtained by solving:

$$\min_{x' \in \mathcal{X}^+_{t_p, t_d}} \text{dist}_G(x, x'), \tag{18}$$

where $\mathcal{X}^+_{t_p, t_d}$ denotes positively classified nodes satisfying confidence and density thresholds. FACE can additionally enforce actionability constraints through condition functions that remove invalid transitions.

## D.3. Evaluation Metrics

We provide the formal definitions of the evaluation metrics used in Section 6.1.

**Validity.**    Given a set of instances $\{x_i\}_{i=1}^n$ and their corresponding recourse solutions $\{x_i'\}_{i=1}^n$, validity is defined as

$$\text{Validity} = \frac{1}{n} \sum_{i=1}^n \mathbf{1}\big[f_{\mathcal{C}}(x_i') = y^+\big]. \tag{19}$$

**Average Cost.**    Let $\mathcal{J}_c$ denote continuous features and $\mathcal{J}_d$ categorical features. We compute the recourse cost as

$$c(x, x') = \sum_{j \in \mathcal{J}_c} \frac{|x_j - x_j'|}{\sigma_j} + \sum_{j \in \mathcal{J}_d} \mathbf{1}[x_j \neq x_j'], \tag{20}$$

where $\sigma_j$ is the standard deviation used for feature normalization. Avg Cost is then the mean of $c(x_i, x_i')$ over all instances.

**Feature Concentration.**    Let $p_{\text{sel}} \in \Delta^d$ denote the learned feature sampling distribution. We measure feature concentration as the effective number of active features:

$$\text{FC} = \exp\big(H(p_{\text{sel}})\big), \quad H(p) = -\sum_{j=1}^d p(j) \log p(j). \tag{21}$$

Smaller values indicate that the search scope is concentrated on fewer features.

## D.4. Implementation Details

This section provides additional implementation details for reproducing ASR-ICL.

### D.4.1. RECOURSE OBJECTIVE AND FEASIBILITY CONSTRAINTS

All recourse solutions are generated by minimizing the objective in Eq. (9):

$$L(x, x') = (f_{\mathcal{C}}(x'))^2 + \lambda\, c(x, x'),$$

subject to feasibility constraints $x' \in \Omega(x)$. We enforce $\Omega(x)$ through a feature-wise projection operator $\Pi_\Omega$, which preserves immutable features and enforces monotonic constraints when specified. Immutable attributes are not allowed to change, while monotonic features can only increase or decrease.

**Australian Credit.** Since all features are anonymized (A1–A14), we treat them as fully actionable and apply no additional feasibility constraints.

**COMPAS.** We treat `is_male` as an immutable attribute and do not impose additional monotonic constraints.

**Diabetes.** We treat `age` and `Pregnancies` as immutable features.

**Corporate Rating.** Since the features correspond to business and financial indicators without natural immutability, we assume all attributes are mutable.

**Student Performance.** We treat `gender`, `age`, and `Ethnicity` as immutable, and restrict `ParentalEducation` to only increase.

### D.4.2. QUERY BUDGET AND OPTIMIZATION SETTINGS

Since $f_{\mathcal{C}}$ is accessed through expensive black-box queries, ASR-ICL is evaluated under a fixed total query budget $B$ for each instance. At each iteration, we sample a subspace of size $k$ and solve the corresponding low-dimensional problem using a RACOS-based zeroth-order optimizer. We use early stopping only as an efficiency heuristic: once a valid recourse is found, we terminate if its cost is within a small margin of the best feasible cost observed so far (e.g., within $1.05\times$). All methods are still compared under the same maximum budget $B$; early stopping only avoids unnecessary additional queries once a satisfactory recourse is found.

*Table 3.* Default hyperparameters used for ASR-ICL.

| Hyperparameter | Value |
|---|---|
| Subspace size $k$ | $\min(5, \lceil \sqrt{d} \rceil)$ |
| Smoothing factor $\alpha$ | 0.5 |
| Total query budget $B$ | 150 |
| Zeroth-order solver | RACOS |
| Feature sampling initialization | Uniform ($I_j = 0$) |

### D.4.3. MIXED FEATURE TYPES

Continuous features are optimized over bounded intervals determined by the empirical minimum and maximum values in the dataset. Categorical features are optimized over discrete grids corresponding to their observed category values. RACOS naturally supports such mixed continuous-discrete search spaces without relaxation.

### D.4.4. IN-CONTEXT PREDICTION SETUP

For each dataset, tabular inputs are formatted as prompts containing $m$ labeled demonstrations sampled uniformly from the training set. All general-purpose LLMs are queried with greedy decoding (temperature $= 0$), and predictions are parsed to obtain class labels.

### D.4.5. RECOURSE EVALUATION PIPELINE

For each dataset, we first obtain predictions from the model $f_{\mathcal{C}}$ on all test instances. Recourse is then generated only for individuals receiving an unfavorable outcome, i.e., those for which $f_{\mathcal{C}}(x) \neq y^+$, where $y^+$ denotes the desired target label defined in Section 3. In binary classification tasks, $y^+$ corresponds to the favorable class. In multi-class datasets, we fix a reference favorable class as the recourse target. For each selected instance, ASR-ICL produces a feasible modification $x'$ that aims to flip the prediction to $y^+$ while minimizing the recourse cost.

## E. Prompt Design

Figure 4 shows the prompt template used for tabular in-context prediction. Each query consists of an instruction header, a set of $m$ labeled demonstrations, and the query instance with the label left blank for completion. The model is required to

---

**Instruction Prompt.**
The task is to provide your best estimate for *Label*. Answer with only one of the candidate class labels. Provide the label only, without any additional text.

**Context.**

> Feature$_1$: value
> Feature$_2$: value
> $\vdots$
> Feature$_d$: value
> Label: $y_i$

**Query.**

> Feature$_1$: value
> Feature$_2$: value
> $\vdots$
> Feature$_d$: value
> Label:

---

*Figure 4.* Prompt template used for model. Each query consists of an instruction prompt, $m$ labeled demonstrations, and a query instance.

output exactly one class label.

## F. Additional Experimental Results

### F.1. Budget and Query Efficiency

Table 4 provides complete recourse validity, cost, and query budget statistics across all evaluated ICL predictors on one representative binary dataset (Australian Credit) and one multi-class dataset (Corporate Rating). Consistent with the main findings, ASR-ICL achieves comparable recourse validity while requiring substantially fewer model queries than the full-space zeroth-order baseline. On the binary task, ASR-ICL typically finds feasible recourse within 20–70 queries, compared to the full budget of 150 queries for Full ZO. Importantly, ASR-ICL also yields significantly lower recourse costs (around 3–5) than the full-space baseline (around 12–14), indicating that the adaptive subspace search promotes more actionable and sparse modifications. On the multi-class task, ASR-ICL similarly reduces query budgets, requiring roughly 40–120 queries across models, while the full-space baseline consistently reaches the maximum budget limit. Recourse costs are again substantially smaller under ASR-ICL, remaining in the range of 3–6 across predictors. These results further confirm that ASR-ICL improves both efficiency and actionability under black-box ICL predictors.

### F.2. Feature Concentration Analysis

We further analyze how ASR-ICL concentrates its search on a small subset of influential features. Recall that ASR-ICL maintains a learned feature selection distribution $p_{\text{sel}}$ over mutable attributes. We measure feature concentration using $\exp(H(p_{\text{sel}}))$, where lower values indicate that the sampler focuses on fewer features. Figures 5 and 6 report the final concentration values at termination under a fixed 32-shot setting. Across both binary and multi-class tasks, ASR-ICL consistently produces substantially more concentrated feature distributions than full-space zeroth-order optimization, supporting the intended adaptive subspace mechanism.

### F.3. Sensitive Analysis

**Sensitivity to the trade-off parameter $\lambda$.** Figure 7 reports the effect of the trade-off parameter $\lambda$ in Eq. (9), which balances validity and proximity. Across a wide range of values ($10^{-3}$ to 0.5), ASR-ICL maintains consistently high recourse validity for most ICL predictors, with only minor fluctuations. At the same time, the average recourse cost varies smoothly and remains within a narrow band, indicating that the method does not require delicate tuning of $\lambda$. In all experiments, we therefore fix $\lambda = 0.1$ as a default choice, which provides a strong validity–cost trade-off across models.

*Table 4.* Query budget statistics (mean ± std) for ASR-ICL and the full-space zeroth-order baseline (Full ZO) on one binary (Australian Credit) and one multi-class (Corporate Rating) dataset.

| Method | Model | Australian Credit | | | Corporate Rating | | |
|---|---|---|---|---|---|---|---|
| | | Validity | Avg Cost | Budget | Validity | Avg Cost | Budget |
| Full ZO | TabPFN-2.5 | $1.00 \pm 0.00$ | $12.88 \pm 0.20$ | $144.28 \pm 14.23$ | $1.00 \pm 0.00$ | $15.44 \pm 0.69$ | $149.45 \pm 0.95$ |
| | TabICL | $1.00 \pm 0.00$ | $12.56 \pm 0.44$ | $150.00 \pm 0.00$ | $0.99 \pm 0.02$ | $18.65 \pm 1.37$ | $150.00 \pm 0.00$ |
| | Qwen3-4B-Instruct | $1.00 \pm 0.00$ | $12.03 \pm 0.22$ | $150.00 \pm 0.00$ | $0.95 \pm 0.06$ | $17.42 \pm 1.70$ | $150.00 \pm 0.00$ |
| | Qwen3-8B | $1.00 \pm 0.00$ | $12.94 \pm 0.97$ | $137.67 \pm 10.79$ | $0.53 \pm 0.11$ | $20.89 \pm 0.55$ | $150.00 \pm 0.00$ |
| | LLaMA-3.2-3B | $1.00 \pm 0.00$ | $14.06 \pm 1.72$ | $150.00 \pm 0.00$ | $0.72 \pm 0.02$ | $18.67 \pm 0.38$ | $150.00 \pm 0.00$ |
| | LLaMA-3.1-8B | $1.00 \pm 0.00$ | $13.00 \pm 0.33$ | $150.00 \pm 0.00$ | $0.26 \pm 0.45$ | $18.73 \pm 0.58$ | $150.00 \pm 0.00$ |
| | GPT-4o | $0.97 \pm 0.04$ | $11.82 \pm 0.71$ | $150.00 \pm 0.00$ | $0.77 \pm 0.03$ | $14.18 \pm 0.57$ | $150.00 \pm 0.00$ |
| ASR-ICL | TabPFN-2.5 | $1.00 \pm 0.00$ | $3.83 \pm 0.04$ | $27.01 \pm 4.85$ | $0.98 \pm 0.02$ | $4.79 \pm 0.19$ | $111.71 \pm 15.85$ |
| | TabICL | $1.00 \pm 0.00$ | $4.47 \pm 0.15$ | $36.29 \pm 10.11$ | $0.99 \pm 0.02$ | $5.79 \pm 0.25$ | $104.60 \pm 10.75$ |
| | Qwen3-4B-Instruct | $0.79 \pm 0.15$ | $3.01 \pm 0.06$ | $56.01 \pm 13.11$ | $0.94 \pm 0.04$ | $3.54 \pm 0.06$ | $117.32 \pm 10.34$ |
| | Qwen3-8B | $0.87 \pm 0.08$ | $2.94 \pm 0.15$ | $44.71 \pm 8.04$ | $0.73 \pm 0.04$ | $3.55 \pm 0.09$ | $105.00 \pm 18.09$ |
| | LLaMA-3.2-3B | $1.00 \pm 0.00$ | $2.99 \pm 0.07$ | $72.38 \pm 10.33$ | $0.95 \pm 0.07$ | $3.01 \pm 0.10$ | $84.44 \pm 9.49$ |
| | LLaMA-3.1-8B | $0.80 \pm 0.08$ | $2.96 \pm 0.05$ | $61.70 \pm 19.70$ | $0.50 \pm 0.24$ | $3.15 \pm 0.11$ | $69.04 \pm 6.65$ |
| | GPT-4o | $0.99 \pm 0.03$ | $4.75 \pm 0.09$ | $49.04 \pm 1.18$ | $0.72 \pm 0.07$ | $4.87 \pm 0.18$ | $40.29 \pm 19.40$ |

*Table 5.* Robustness to context distribution and ordering under the fixed 32-shot setting.

| Dataset | Condition | Demo Dist. | TabPFN Validity | TabPFN Cost | Qwen Validity | Qwen Cost |
|---|---|---|---|---|---|---|
| COMPAS | balanced | {0:16, 1:16} | 1.00 | 3.15 | 1.00 | 2.99 |
| COMPAS | imbal_0dom | {0:24, 1:8} | 1.00 | 3.15 | 1.00 | 3.06 |
| COMPAS | imbal_1dom | {1:24, 0:8} | 1.00 | 3.11 | 1.00 | 3.37 |
| COMPAS | sorted_0first | {0:16, 1:16} | 1.00 | 3.18 | 1.00 | 3.05 |
| COMPAS | sorted_1first | {1:16, 0:16} | 1.00 | 3.20 | 0.50 | 3.28 |
| Australian Credit | balanced | {0:16, 1:16} | 1.00 | 3.47 | 0.79 | 3.61 |
| Australian Credit | imbal_0dom | {0:24, 1:8} | 0.81 | 3.92 | 0.70 | 3.38 |
| Australian Credit | imbal_1dom | {1:24, 0:8} | 1.00 | 3.58 | 0.77 | 3.45 |
| Australian Credit | sorted_0first | {0:16, 1:16} | 0.94 | 3.44 | 0.79 | 3.47 |
| Australian Credit | sorted_1first | {1:16, 0:16} | 1.00 | 3.68 | 0.70 | 3.54 |

**Sensitivity to subspace size $k$.** Figure 8 evaluates the effect of the subspace dimension $k$ in ASR-ICL on a representative models (Qwen3-4B-Instruct). When $k$ is very small, recourse validity is noticeably reduced, since too few actionable features are available to construct a successful modification. As $k$ increases, validity rapidly improves and stabilizes around $k = 3$–$5$, indicating that only a small number of features is typically sufficient for recourse. At the same time, the average cost grows with larger $k$, reflecting the increased flexibility of modifying more features and the higher complexity of the inner zeroth-order search. Overall, these results suggest that moderate subspace sizes provide the best validity–efficiency trade-off. Accordingly, we adopt the default choice $k = \min(5, \lceil\sqrt{d}\rceil)$ throughout our experiments, which consistently yields stable performance without tuning.

### F.4. Robustness Analysis

We conduct additional experiments under a fixed 32-shot setting to evaluate robustness to context distribution and ordering (as shown in Table 5). Overall, ASR-ICL remains stable across most settings. For TabPFN, validity is consistently high, with only moderate degradation under strong class imbalance. For LLM-based models (e.g., Qwen3-4B-Instruct), we observe higher sensitivity to ordering in some cases (e.g., validity drops under sorted contexts), indicating model-dependent prompt sensitivity. These results suggest that while context composition can influence the induced decision rule, ASR-ICL remains robust under a wide range of realistic conditions.

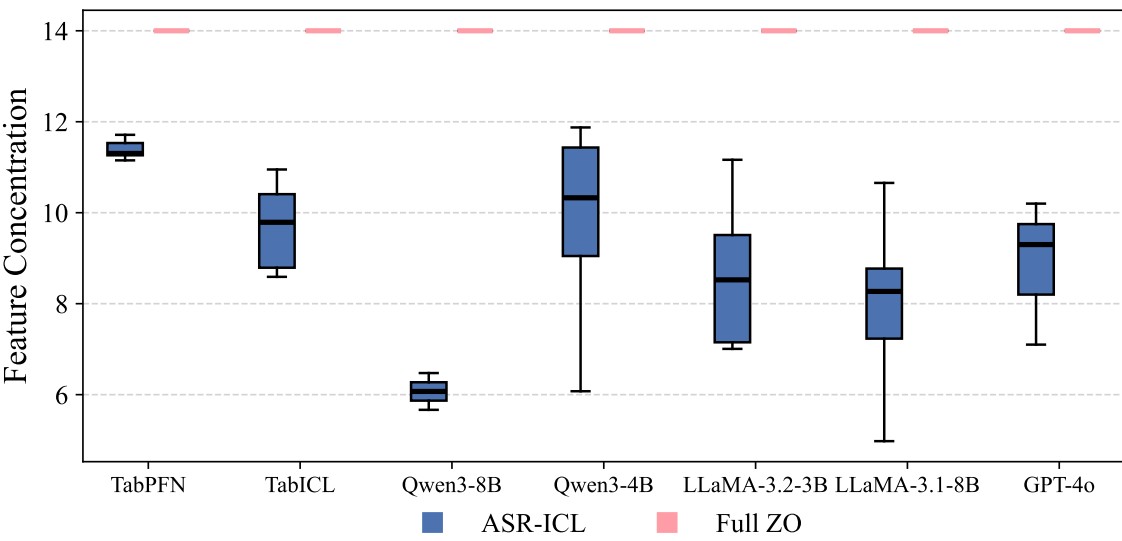

*Figure 5.* Final feature concentration on Australian Credit (32 shots). ASR-ICL yields lower effective feature counts, indicating stronger focus on a small subset of important feature compared to Full ZO.

*Table 6.* Sensitivity to target class in multi-class recourse on Corporate Rating.

| Model | Target | Validity | Cost |
|---|---|---|---|
| | 0 | $0.87 \pm 0.04$ | $5.00 \pm 0.16$ |
| TabPFN-2.5 | 1 | $0.95 \pm 0.02$ | $5.04 \pm 0.11$ |
| | 2 | $0.96 \pm 0.02$ | $4.90 \pm 0.21$ |
| | 0 | $0.58 \pm 0.18$ | $3.74 \pm 0.53$ |
| Qwen3-4B-Instruct | 1 | $0.85 \pm 0.13$ | $3.80 \pm 0.07$ |
| | 2 | $0.91 \pm 0.06$ | $3.87 \pm 0.13$ |

## F.5. Sensitivity to different target in multi-class recourse

We run additional experiments by varying the target class (as shown in Table 6). For TabPFN, both validity and cost are stable across targets. For Qwen3-4B-Instruct, validity varies more (0.58–0.91), while cost remains largely unchanged. This suggests that the variation mainly reflects differences in target difficulty rather than instability of the method.

## G. Case Study

Table 7 presents a representative recourse example on a real instance from the Diabetes dataset. In contrast, ASR-ICL achieves recourse by modifying only two key actionable attributes (Glucose and BMI), yielding a sparser and more interpretable recommendation.

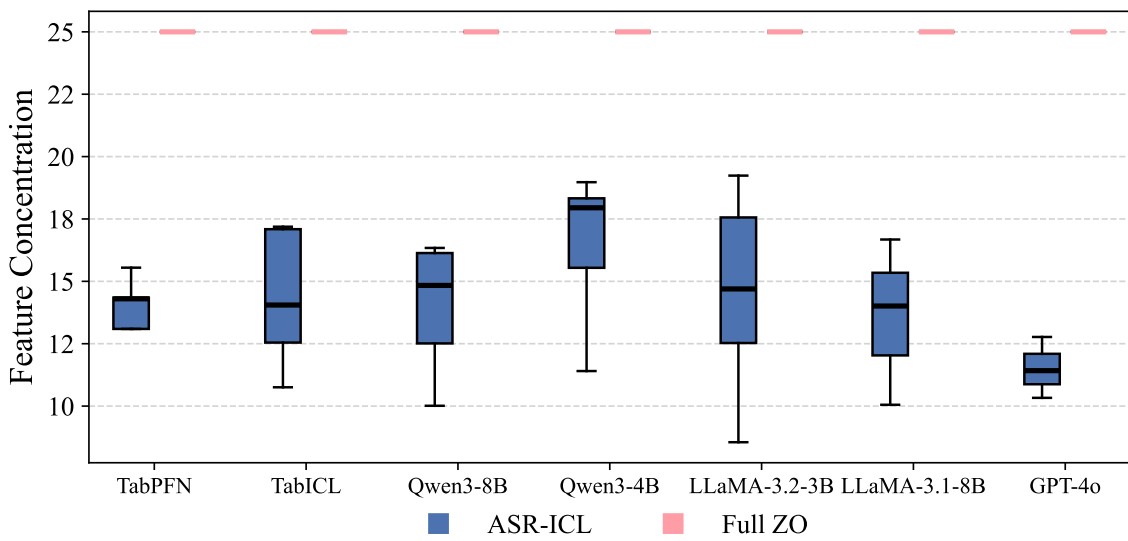

*Figure 6.* Final feature concentration on Corporate Rating (48 shots). ASR-ICL remains consistently more focused than full-space baselines, demonstrating that adaptive subspace search extends naturally beyond binary settings.

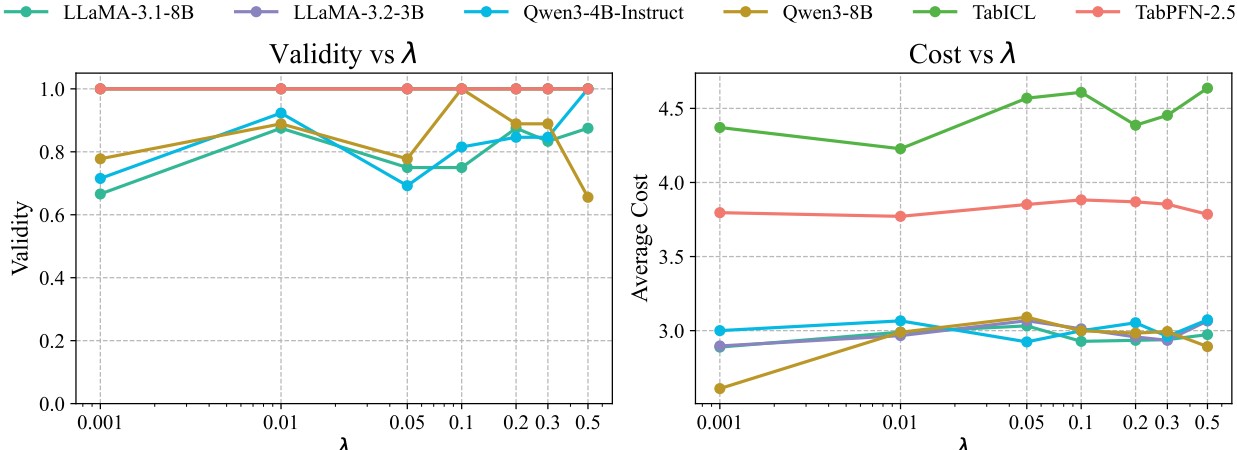

*Figure 7.* Sensitivity of ASR-ICL to the trade-off parameter $\lambda$ on Australian Credit across different models.

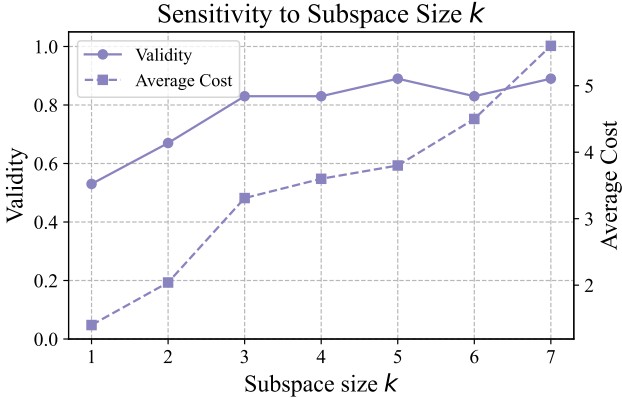

*Figure 8.* Sensitivity of ASR-ICL to the subspace size $k$ on a representative model.

*Table 7.* Case study on a real instance from the Diabetes dataset (Outcome=1). We compare recourse suggestions from classical baselines, Full ZO and ASR-ICL. Bold entries indicate modified features. We treat *Age* and *Pregnancies* as immutable features.

| Feature | Original | DiCE | AR | FACE | Full ZO | ASR-ICL |
|---|---|---|---|---|---|---|
| Pregnancies | 6 | **3** | 6 | 6 | 6 | 6 |
| Glucose | 148 | **138** | **136** | **134** | **135** | **130** |
| BloodPressure | 72 | **69** | 72 | **68** | **68** | 72 |
| SkinThickness | 35 | **33** | 35 | **30** | **28** | 35 |
| Insulin | 0 | 0 | 0 | **60** | **60** | 0 |
| BMI | 33.6 | **33.2** | **31.0** | **30.5** | **30.8** | **30.5** |
| DiabetesPedigreeFunction | 0.627 | 0.627 | 0.627 | 0.627 | **0.672** | 0.627 |
| Age | 50 | **35** | 50 | 50 | 50 | 50 |
| Changed features | – | 5 | 2 | 5 | 6 | 2 |
| Actionable | – | No | Yes | Yes | Yes | Yes |
| Outcome flipped | – | Yes | Yes | Yes | Yes | Yes |

