# OpenReview forum: "Algorithmic Recourse of In-Context Learning for Tabular Data"
_ICML.cc/2026/Conference — ICML 2026 regular_

### Official Review · Reviewer_df7r · 2026-02-27

**Soundness:** 3
**Presentation:** 2
**Significance:** 2
**Originality:** 3
**Overall Recommendation:** 4
**Confidence:** 3

**Summary:**

This paper systematically investigates recourse in in-context learning (ICL) for tabular data. A theoretical analysis of recourse under the ICL framework is first presented, characterizing its feasibility conditions and key influencing factors. Building upon this analysis, a subspace-based ICL recourse method tailored to tabular data is proposed to alleviate the optimization challenges arising from high-dimensional feature spaces. By performing optimization within low-dimensional subspaces, the approach effectively reduces computational complexity. Experimental results on ICL models demonstrate the effectiveness of the proposed method.

**Compliance With Llm Reviewing Policy:**

Affirmed.

**Final Justification:**

The paper explores an interesting problem, and I appreciate the authors’ response. However, I still believe the manuscript falls short in articulating why recourse to ICL methods is truly necessary and important. Since this point is central to the paper’s motivation and practical significance, I view it as a substantive weakness rather than a minor writing issue. Therefore, although I see potential in the idea, I am not convinced the current version meets the bar for acceptance.

**Key Questions For Authors:**

Mainly see weaknesses.

In summary, the core issues are:

1 (major). The motivation and the necessity of designing the method specifically for ICL are not sufficiently justified.
2. The trade-off between recourse feasibility and predictive performance lacks in-depth analysis.
3 (major). The time and memory overhead of the algorithm are not discussed.
4 (major). The meanings of key notations are unclear and potentially confusing.


The score will be further adjusted based on the authors’ responses to the above concerns.

**Limitations:**

The authors do not explicitly discuss these limitations in the paper. It is recommended that a dedicated discussion section be included to address the weaknesses of the proposed method, particularly regarding predictive performance trade-offs and computational overhead.

**Strengths And Weaknesses:**

## Strengths

1. The paper integrates both theoretical analysis and empirical evaluation, providing theoretical feasibility guarantees for recourse in tabular ICL and strengthening the foundation of this research direction.
2. The problem addressed is practically meaningful in real-world tabular learning scenarios, enhancing the applied value of the work.

## Weaknesses

1. The motivation requires further clarification. Although recourse is an important problem, the importance and necessity of studying recourse specifically under the ICL framework should be more thoroughly justified. Moreover, the proposed subspace exploration strategy appears applicable to non-ICL recourse methods as well. However, the paper claims that the method is specifically designed for ICL. The authors should clearly articulate the unique aspects of recourse in ICL settings and explain why the proposed approach is particularly suitable for ICL or why it may not directly apply to non-ICL models.

2. The algorithmic design needs further discussion. According to Proposition 4.1, if at least one unbounded feature exists in tabular data, selecting that feature as a sub-dataset for recourse optimization would theoretically guarantee feasibility. However, two issues arise:

   * First, such unbounded features are often difficult to identify or may not exist in practical tabular datasets.
   * Second, when only a small subset of features is selected as the optimization subspace, predictive performance is likely to degrade, resulting in a performance–recourse trade-off.

   Therefore, beyond recourse validity, it is important to analyze the potential performance loss under restricted subspaces. In addition, a comparison between (i) the strategy of selecting at least one unbounded feature and (ii) the proposed subspace feature selection method should be conducted in terms of recourse validity, in order to better demonstrate the effectiveness and necessity of the proposed design.

3. The computational cost of the algorithm is not discussed. Compared with traditional models, inference with large language models or TabPFN-style predictors typically incurs higher time and memory overhead. An analysis of runtime and resource consumption would help readers and practitioners make informed decisions when selecting recourse methods.

4. The presentation of the loss function $L$ is confusing. In Lines 161, 271, and 223 (right column), three seemingly different formulations of $L$ are provided. A similar inconsistency appears for $f_C$: it is unclear whether it represents predicted labels or prediction scores. The interpretation of its values is also ambiguous, particularly since the signs in Lines 161 and 271 appear inconsistent (even opposite). It should be clarified whether larger values are better or worse. Such core notation and sign conventions must be unified throughout the paper. Furthermore, it is unclear whether Algorithm 1 uses the loss defined in Equation (9) or the indicator function described in Line 223 (right column), as the descriptions appear inconsistent. Clear and consistent notation is necessary to avoid confusion.

---

> ### Author Rebuttal · Authors · 2026-03-30
>
> Thank you for your insightful comments and suggestions. Here is our response.
>
> >Response to W1:
>
> The key motivation for studying recourse under ICL is that the prediction function is no longer fixed, but depends on the input-dependent context. This fundamentally changes the recourse problem: the decision boundary varies across instances, and standard recourse methods that assume a fixed trained model are not directly applicable.  In addition, ICL-based predictors are typically accessible only via expensive black-box queries, without gradients or model parameters. This makes recourse a query-limited black-box optimization problem. Our proposed strategy is generally applicable to black-box recourse problems, including non-ICL models. However, it is particularly well-suited to the ICL setting, where gradient information is unavailable and query efficiency is critical. In contrast, classical recourse methods can often leverage gradients or model structure, reducing the need for such strategies. We will clarify this positioning in the revision.
>
> >Response to W2:
>
> We thank the reviewer for these comments. We clarify that Proposition 4.1 provides an existence result under idealized assumptions (e.g., unbounded features), and is not intended as a practical recourse strategy. In practice, features are typically bounded and constrained, so feasibility must be achieved through optimization rather than trivial feature changes. We agree that restricting optimization to a small subspace introduces a trade-off. This is analyzed in Fig. 8, where very small k reduces validity, while moderate k quickly recovers high validity with low cost. To isolate the effect of feature selection, we further compare ASR-ICL with a random-subspace baseline under the same setting (T1). Random selection leads to significantly lower validity and higher query cost, while ASR-ICL remains both effective and efficient. These results show that the improvement comes from adaptive subspace selection, rather than dimensionality reduction alone.
>
> **T1: Ablation on adaptive vs. random feature selection.**
> | Model             | Method   | Australian Credit (validity) | cost        | Budget        | Corporate Rating (validity) | cost        | Budget        |
> |------------------|----------|------------------------------|-------------|---------------|-----------------------------|-------------|---------------|
> | TabPFN-2.5       | random   | 0.82 ± 0.02                 | 3.49 ± 0.27 | 139.14 ± 7.32 | 0.72 ± 0.06                | 3.81 ± 0.13 | 150.00 ± 0.00 |
> |                  | adaptive | 1.00 ± 0.00                 | 3.83 ± 0.04 | 27.01 ± 4.85  | 0.98 ± 0.02                | 4.79 ± 0.19 | 111.71 ± 15.85|
> | Qwen-4B-instruct | random   | 0.45 ± 0.20                 | 3.42 ± 0.03 | 150.00 ± 0.00 | 0.59 ± 0.17                | 3.76 ± 0.06 | 150.00 ± 0.00 |
> |                  | adaptive | 0.79 ± 0.15                 | 3.01 ± 0.06 | 56.01 ± 13.11 | 0.94 ± 0.04                | 3.54 ± 0.06 | 117.32 ± 10.34|
>
> >Response to W3:
>
> In our setting, the dominant computational cost comes from querying the underlying model (e.g., LLMs or TabPFN), which is significantly more expensive than the optimization steps of the recourse algorithm itself. Therefore, we evaluate efficiency in terms of query complexity, which directly determines runtime and resource usage in practice. As shown in our experiments, ASR-ICL reduces the number of queries by approximately 2–3× compared to full-space zeroth-order optimization, leading to proportionally lower runtime and API cost when using large models. We will clarify this discussion in the revision and better relate query complexity to practical runtime considerations.
>
> >Response to W4:
>
> We agree that the current presentation of the loss function and the notation for $ f_C $ can be confusing. The different formulations serve distinct purposes: (i) Eq. (6) is a smooth surrogate used in the theoretical analysis, where $ f_C(x) $ denotes a continuous score; (ii) Eq. (9) is the practical objective used in the black-box setting; (iii) the indicator function is used only for evaluating recourse success.
>
> We will revise the paper to clearly separate these roles, unify notation, and make sign conventions consistent throughout. We will also clarify that Algorithm 1 optimizes Eq. (9), while the indicator is used only for evaluation.

---

> > ### Author Rebuttal · Reviewer_df7r · 2026-04-01
> >
> > Update after second round rebuttal
> >
> > I appreciate the authors’ response. My concerns have been addressed.

---

> > > ### Author Response · Authors · 2026-04-01
> > >
> > > We thank the reviewer for the comments.
> > >
> > > >Response to W1:
> > >
> > > We agree that ICL models may have limitations in large-scale prediction tasks. Our goal, however, is not to argue that ICL is universally optimal, but to study recourse in settings where in-context model is already being actively explored. Recent work shows that ICL-based models can effectively handle tabular prediction tasks without task-specific training [1,2,3], and are increasingly studied as general-purpose models. A concrete scenario where this arises is systems that rely on in-context models without retraining, such as few-shot tabular prediction and emerging ICL-based decision-making settings in domains such as finance [4]. In such settings, users interact directly with a context-conditioned model, whose effective decision rule depends on the provided context. Even when the underlying parameters remain fixed, changing the context changes the effective predictor seen by the user. Therefore, recourse must be defined with respect to the context-conditioned decision rule rather than a single global model. Importantly, our method only requires black-box access and does not rely on model internals. As a result, it is directly applicable not only to current ICL models, but also to future models with improved predictive performance. Our work focuses on this setting, where recourse must operate under context-dependent, black-box models. We will clarify this motivation in the revision.
> > >
> > >
> > > [1] TabPFN-2.5: Advancing the State of the Art in Tabular Foundation Models
> > >
> > > [2] ConTextTab: A Semantics-Aware Tabular In-Context Learner
> > >
> > > [3] TabICL: A Tabular Foundation Model for In-Context Learning on Large Data
> > >
> > > [4] Breaking the Gradient Barrier: Unveiling Large Language Models for Strategic Classification
> > >
> > > >Response to W2:
> > >
> > > The performance of using all features (i.e., full-space optimization) is already included in our paper via the Full ZO baseline. Under the same black-box access and query budget, Full ZO serves as a natural upper-bound baseline, as it searches over the entire feature space without subspace restriction. While it is not a theoretical optimum, it represents the strongest practical baseline under the same black-box access and query constraints. Compared to Full ZO, our method achieves comparable or higher validity while requiring significantly fewer queries (Figure 3, Table 4, Table 5). This indicates that the proposed strategy does not sacrifice recourse quality, but improves efficiency. We will clarify this connection in the revision and explicitly present Full ZO as the upper-bound baseline.

---

### Official Review · Reviewer_eQEr · 2026-03-07

**Soundness:** 3
**Presentation:** 2
**Significance:** 2
**Originality:** 3
**Overall Recommendation:** 4
**Confidence:** 2

**Summary:**

This paper studies algorithmic recourse for tabular prediction under in-context learning. The paper gives a theoretical analysis in a linear self-attention setting, and proposes ASR-ICL, a black-box zeroth-order method that searches for recourse in adaptive low-dimensional feature subspaces.

**Compliance With Llm Reviewing Policy:**

Affirmed.

**Final Justification:**

Thanks to the author for responding to all my concerns and follow-up questions, I would like to hold my assessment.

**Key Questions For Authors:**

- In Section 4 the theory is developed for a linear regression problem with a single-layer linear self-attention model, while Section 6.1 evaluates black-box classification systems such as TabPFN-2.5, TabICL, Qwen, LLaMA, and GPT-4o. Which parts of the empirical story do the authors believe are directly supported by the theory, and which parts should be viewed as heuristic transfer only? This would affect my confidence in the paper’s main claims.
- In the multi-class setting described in Appendix D.4.5, the paper fixes a favorable target class as the recourse target. How sensitive are the results to this choice? It would be helpful to know whether validity or cost changes a lot for different target classes.

**Limitations:**

I think the paper should discuss its limitations more clearly, especially the gap between the theory and the practical models, and the dependence of recourse on how the context is built.

**Strengths And Weaknesses:**

### Strengths：
- This paper studies a new problem. Most prior recourse work assumes a fixed trained model, while this paper looks at the ICL setting where predictions depend on the chosen context.
- The paper explains why recourse under ICL is not a trivial extension of standard recourse.
- The experimental section is also fairly broad, covering specialized tabular ICL models, general LLMs, binary tasks, multi-class tasks, and some analysis of context size and query efficiency.

### Weaknesses:
- The theory in Section 4 is developed for a linear regression problem with a single-layer linear self-attention model trained by gradient flow, while the experiments use black-box classification systems such as TabPFN-2.5, Qwen, and GPT-4o. The theory is interesting, but the paper should be more careful about which conclusions are formally supported and which are only suggested by analogy.
- The paper studies the number of context examples and simple context resampling, but the paper does not give a more systematic analysis of demonstration selection, ordering, or retrieval policy. Since context dependence is the main reason this setting is difficult, I would have liked more evidence on robustness to these choices.
- There are several minor writing and formatting issues that should be cleaned up in the final version. For example,
   - Section 4.1 contains a duplicated word (“the the”; line 415).
   - Section 6.3 also includes a few small grammar issues, for instance “model behave” instead of “models behave” or “the model behaves,” the misspelling “diffeent.”

   These are all minor copy-editing issues and do not affect the main technical content, but fixing them would improve the polish and professionalism of the paper.

---

> ### Author Rebuttal · Authors · 2026-03-30
>
> Thanks for your comments and suggestions.
>
> >Response to W1 and Q1:
>
> We agree that our theoretical results are formally established under a simplified linear setting, while the experiments involve more complex black-box models. Our intention is to provide mechanism-level insights rather than a complete characterization of modern ICL systems. Specifically, the theory formally supports: (i) the well-definedness of recourse under context-conditioned predictors, (ii) boundedness and feasibility guarantees, and (iii) convergence toward classical recourse as context size increases. In contrast, the empirical results extend these insights to modern ICL models, including: (i) improved stability and reduced cost with larger context size, and (ii) the effectiveness of subspace-based optimization in high-dimensional black-box settings. These should be interpreted as empirical validation of the qualitative predictions, rather than formal guarantees. Importantly, our experimental design is directly aligned with these theoretical predictions (e.g., varying context size and studying high-dimensional optimization), allowing us to test whether the same qualitative behaviors emerge in practice.
>
> >Response to W2:
>
> T1: Robustness to context distribution and ordering (32-shot).
>
> | Dataset           | Condition       | Demo Dist       | TabPFN Validity | TabPFN Cost  | Qwen3-4B-Instruct Validity | Qwen3-4B-Instruct Cost  |
> |-------------------|-----------------|-----------------|-----------------|---------------|---------------|-------------|
> | COMPAS            | balanced        | {0:16, 1:16}    | 1.00            | 3.15          | 1.00          | 2.99        |
> | COMPAS            | imbal_0dom      | {0:24, 1:8}     | 1.00            | 3.15          | 1.00          | 3.06        |
> | COMPAS            | imbal_1dom      | {1:24, 0:8}     | 1.00            | 3.11          | 1.00          | 3.37        |
> | COMPAS            | sorted_0first   | {0:16, 1:16}    | 1.00            | 3.18          | 1.00          | 3.05        |
> | COMPAS            | sorted_1first   | {1:16, 0:16}    | 1.00            | 3.20          | 0.50          | 3.28        |
> | Australian Credit | balanced        | {0:16, 1:16}    | 1.00            | 3.47          | 0.79          | 3.61        |
> | Australian Credit | imbal_0dom      | {0:24, 1:8}     | 0.81            | 3.92          | 0.70          | 3.38        |
> | Australian Credit | imbal_1dom      | {1:24, 0:8}     | 1.00            | 3.58          | 0.77          | 3.45        |
> | Australian Credit | sorted_0first   | {0:16, 1:16}    | 0.94            | 3.44          | 0.79          | 3.47        |
> | Australian Credit | sorted_1first   | {1:16, 0:16}    | 1.00            | 3.68          | 0.70          | 3.54        |
>
> We conduct additional experiments under a fixed 32-shot setting to evaluate robustness to context distribution and ordering. Overall, ASR-ICL remains stable across most settings. For TabPFN, validity is consistently high, with only moderate degradation under strong class imbalance. For LLM-based models (e.g., Qwen), we observe higher sensitivity to ordering in some cases (e.g., validity drops under sorted contexts), indicating model-dependent prompt sensitivity. These results suggest that while context composition can influence the induced decision rule, ASR-ICL remains robust under a wide range of realistic conditions. We will include these results in the revision.
>
> >Response to W3:
>
> We thank the reviewer for pointing out these writing and formatting issues. We will carefully proofread the paper and fix all typos and grammar errors in the final version to improve clarity and presentation.
>
> >Response to Q2:
>
> T2: Sensitivity to target class in multi-class recourse (Corporate Rating).
>
> | Model               | Target | Validity  | Cost  |
> |---------------------|--------|------------|--------|
> | **TabPFN-2.5**      | 0      | 0.87 ± 0.04 | 5.00 ± 0.16 |
> |                     | 1      | 0.95 ± 0.02 | 5.04 ± 0.11 |
> |                     | 2      | 0.96 ± 0.02 | 4.90 ± 0.21 |
> | **Qwen3-4B-Instruct** | 0      | 0.58 ± 0.18 | 3.74 ± 0.53 |
> |                     | 1      | 0.85 ± 0.13 | 3.80 ± 0.07 |
> |                     | 2      | 0.91 ± 0.06 | 3.87 ± 0.13 |
>
> We run additional experiments by varying the target class (T2). For TabPFN, both validity and cost are stable across targets. For Qwen, validity varies more (0.58–0.91), while cost remains largely unchanged. This suggests that the variation mainly reflects differences in target difficulty rather than instability of the method. We will include these results in the revision.

---

> > ### Author Rebuttal · Reviewer_eQEr · 2026-04-02
> >
> > I thank the authors for addressing my concerns. I will keep my score unchanged, which remains positive.

---

### Official Review · Reviewer_ysUk · 2026-03-09

**Soundness:** 2
**Presentation:** 3
**Significance:** 3
**Originality:** 3
**Overall Recommendation:** 4
**Confidence:** 3

**Summary:**

The paper studies algorithmic recourse for tabular data under in-context learning. The author first establishes theoretical foundations for ICL-based recourse, formalizing the concept to prove it is well-defined, bounded and feasible, and deriving a high-probability upper bound for optimal recourse cost. Then, the authors propose Adaptive Subspace Recourse for In-Context Learning (ASR-ICL), a zeroth-order optimization framework tailored for black-box tabular ICL models, which adaptively concentrates optimization on a small subset of influential features via probabilistic subspace sampling and iterative importance scoring. It is worth noting that ASR-ICL framework extends to multi-class tabular tasks. Experiments on real-world binary/multi-class datasets across specialized tabular ICL models, open-source LLMs and GPT-4o demonstrate the performance of ASR-ICL.  it matches the recourse validity of classic baselines for trained models with substantially lower costs, empirically confirms the theoretical convergence, and exhibits robustness to context resampling.

**Compliance With Llm Reviewing Policy:**

Affirmed.

**Final Justification:**

Considering the authors‘ replies. I will keep my positive scores unchanged.

**Key Questions For Authors:**

1. In the theoretical analysis and the proposed method, is the correlation between features considered? In many tabular datasets features are highly correlated, so it would be helpful to clarify or analyze this.
2. For the experiments with LLMs, a simple baseline could be to directly prompt the LLM to suggest recourse actions (i.e., asking the model to propose the recourse that flips the prediction), since LLMs already possess certain domain knowledge and reasoning capabilities. Have the authors considered such a baseline or conducted any comparison or analysis in this direction?

**Limitations:**

yes

**Strengths And Weaknesses:**

strengths

1. The paper studies an important and emerging problem, algorithmic recourse for in-context learning models, and, according to the authors, is the first study of algorithmic recourse for tabular data under ICL, providing useful insights.
2. The paper provides meaningful theoretical analysis, including feasibility and convergence results, a closed-form characterization of recourse in the stylized setting, and a practical ASR-ICL framework.
3. The paper is clearly written and well organized.
4. The experiments cover a diverse set of modern models, including both open-source and closed-source LLMs such as Qwen, LLaMA, and GPT-4o.

weakness

1. The theoretical analysis is limited to a single-layer linear self-attention model, while the actual experiments involve much more complex models such as Qwen3 and GPT-4o.
2. The experimental baselines appear somewhat outdated, with the most recent main comparison methods largely coming from around 2020.
3. For the large-model experiments, the paper mainly demonstrates effectiveness but does not compare against recourse baselines under the same black-box LLM setting.

---

> ### Author Rebuttal · Authors · 2026-03-30
>
> Thank you so much for your insightful comments and suggestions. Here is our response.
>
> >Response to W1:
>
> As discussed with Reviewer ETEX, our theoretical analysis is conducted in a simplified linear setting to provide mechanism-level insights, rather than a direct modeling of modern LLMs. This follows standard practice in recent theoretical work on ICL [1,2,3]. Instead, it is designed to capture the mechanism of recourse under context-dependent predictors. In particular, it shows that recourse remains well-defined and that recourse cost decreases and stabilizes as the number of context examples increases. We empirically validate these behaviors in Sec. 6.3 across different ICL models (TabPFN, TabICL, Qwen, GPT-4o), where we observe consistent trends despite architectural differences. We will clarify this positioning and the connection between theory and experiments in the revision.
>
> [1] Transformers learn in-context by gradient descent
>
> [2] Trained Transformers Learn Linear Models In-Context
>
> [3] Transformers learn to implement preconditioned gradient descent for in-context learning
>
> >Response to W2, W3 and Q2:
>
> **T1: Additional baseline on one binary and one multi-class dataset.**
> ElliCE is evaluated in its native trained-model setting.
> ASR-ICL and Direct LLM Recourse operate under black-box ICL models.
>
> | Method               | Model                 | Australian Credit |        | Corporate Rating |        |
> |----------------------|------------------------|-------------------|--------|-------------------|--------|
> |                      |                        | Validity         | Cost  | Validity         | Cost  |
> | **Trained Models**   |                        |                   |        |                   |        |
> | ElliCE               | MLP                    | 1.00 ± 0.00       | 13.41 ± 0.12 | – | – |
> | ElliCE               | Logistic Regression    | 1.00 ± 0.00       | 12.00 ± 0.48 | – | – |
> |                      |                        |                   |        |                   |        |
> | **ICL-based Models** |                        |                   |        |                   |        |
> | ASR-ICL              | TabPFN-2.5             | 1.00 ± 0.00       | 3.83 ± 0.04 | 0.98 ± 0.02 | 4.79 ± 0.19 |
> | ASR-ICL              | Qwen3-4B-Instruct      | 0.79 ± 0.15       | 3.01 ± 0.06 | 0.94 ± 0.04 | 3.54 ± 0.06 |
> | Direct LLM Recourse  | TabPFN-2.5             | 0.72 ± 0.09       | 3.38 ± 0.13 | 0.30 ± 0.12 | 5.54 ± 0.11 |
> | Direct LLM Recourse  | Qwen3-4B-Instruct      | 0.73 ± 0.07       | 2.94 ± 0.05 | 0.25 ± 0.09 | 3.68 ± 0.14 |
>
> We conducted additional experiments (T1). We acknowledge that many widely adopted recourse baselines originate around 2020. However, as discussed in our related work, more recent studies primarily focus on robustness, rather than directly optimizing recourse validity and cost. As a result, most existing works still rely on or compare against these earlier baselines [4]. To provide a more up-to-date comparison, we additionally include a recent method, ElliCE [5], in our experiments.
>
> To the best of our knowledge, there are no prior recourse methods designed for black-box ICL models. The most relevant baseline under the same setting is full-space zeroth-order optimization (Full ZO), which shares identical access assumptions (no gradients, query-based) with ASR-ICL. This comparison is already included in the paper (Table 4 / Figure 3), where ASR-ICL achieves comparable or higher validity with substantially fewer queries.
>
> Following the reviewer’s suggestion, we additionally introduce a new baseline, Direct LLM Recourse, where a strong LLM (GPT-4o) is prompted to directly propose feature modifications that flip the prediction.
>
> As shown in T1, Direct LLM Recourse can sometimes produce low-cost suggestions, but fails to reliably achieve valid recourse, especially in more complex settings (e.g., validity drops to 0.25–0.30 on Corporate Rating). This indicates that LLM reasoning alone is insufficient to guarantee prediction flipping under context-conditioned decision rules. In contrast, ASR-ICL consistently achieves high validity while maintaining low cost across both datasets. Moreover, while ElliCE attains perfect validity on trained models, it incurs substantially higher cost (12–13 vs 3–5), and is not directly applicable to black-box ICL models.
>
>  [4] HARE: Human-in-the-Loop Algorithmic Recourse
>
>  [5] ElliCE: Efficient and Provably Robust Algorithmic Recourse via the Rashomon Sets
>
> >Response to Q1:
>
> In the theoretical analysis, feature correlation is explicitly modeled via the covariance matrix, and both the predictor and the optimal recourse solution depend on covariance terms. In ASR-ICL, correlations are handled implicitly: the adaptive subspace selection is data-driven and naturally captures useful feature interactions, while feasibility constraints ensure realistic modifications.

---

> > ### Author Rebuttal · Reviewer_ysUk · 2026-04-04
> >
> > Thanks the authors for replying my concerns. I will keep my positive scores.

---

### Official Review · Reviewer_ETEX · 2026-03-12

**Soundness:** 3
**Presentation:** 4
**Significance:** 2
**Originality:** 3
**Overall Recommendation:** 4
**Confidence:** 1

**Summary:**

The paper present a new problem: "recourse in the case where the predictor is an in-context learning method". Some of the contribution are: (1) develop a theoretical analysis for this simplified case, (2) develop a method for this case, and (3) empirical results that by increasing context more stable prediction are obtained.

**Compliance With Llm Reviewing Policy:**

Affirmed.

**Final Justification:**

The authors have resolved my concern.

**Key Questions For Authors:**

see Weaknesses.

**Limitations:**

yes

**Strengths And Weaknesses:**

## Strengths

* The paper proposes a new and interesting problem of "recourse" under context-based tabular prediction methods. The motivation is clear.
* The paper provides theoretical results.

## Weaknesses (more than weaknesses, concerns)

* I do not completely understand the purpose of Table 1. To my understanding, the methods at the top of the table are recourse methods in the classical setting, i.e., for a trained classifier that is independent of context. The bottom part shows the proposed recourse method using different ICL methods. However, I do not really understand what I should take away from these results, since these "baselines" are not comparable.

* To my understanding, the theory studies the case of a single-layer linear self-attention model in a Gaussian setup, along with some bounds as the context grows. This setting seems a bit far from what is actually reported, since the models used in practice are TabPFN, TabICL, Qwen, LLaMA, and GPT-4o. Why not test the theory in a more synthetic setup, where you can train your own Tab(PFN/ICL) method under these assumptions? Otherwise, the theory feels somewhat disconnected from what is done in practice.

* The paper contains ablation studies, but the central component seems to be the adaptive importance update, which aims to learn which features matter most and then spend more effort searching over those features. However, the experiments do not clearly show whether this part is actually helping. There is no comparison to a simpler version that picks features at random each time, or to one that uses a fixed feature-selection rule instead of updating it throughout the process. Because of that, it is hard to tell whether the improvement really comes from the paper’s new adaptive feature-selection idea, or simply from the overall optimization procedure.

---

> ### Author Rebuttal · Authors · 2026-03-30
>
> Thank you very much for your recognition of our work. Here is our response.
> >Response to W1:
>
> We agree that the methods in Table 1 are not strictly comparable, as they operate under different paradigms. However, the purpose of Table 1 is not to provide a  benchmark, but to answer a more fundamental question: whether recourse under ICL is meaningful in practice, and how its scale compares to classical settings. Specifically, Table 1 provides two key insights: (i) Feasibility: ASR-ICL consistently achieves high validity across models, demonstrating that actionable recourse can indeed be obtained under ICL. (ii) Scale of recourse: The resulting recourse costs are of a similar magnitude to classical methods, indicating that recourse under ICL is not degenerate or prohibitively expensive. Without such a reference, it would be difficult to interpret whether the obtained recourse under ICL is reasonable.
>
> >Response to W2:
>
> We agree that our theoretical analysis is conducted under a simplified setting, while the empirical models are significantly more complex. We note that this follows standard practice in recent theoretical work on in-context learning, which analyzes simplified attention-based models to obtain tractable insights [1,2,3]. While it is possible to construct synthetic models that match the theoretical assumptions, such setups would only verify the theory in a controlled setting and would not reflect the behavior of modern ICL models, which operate as black-box predictors with complex context dependence.
>
> Our goal is not to faithfully model modern architectures, but to capture the mechanism of recourse under context-dependent predictors. In particular, the theory predicts that (i) recourse remains well-defined, (ii) recourse becomes more stable as the number of context examples increases, and (iii) high dimensionality motivates subspace-based optimization. These trends are empirically validated in Sec. 6.3 across diverse ICL models (TabPFN, TabICL, Qwen, LLaMA, GPT-4o), despite their architectural differences.
>
> We will revise the paper to better distinguish formal guarantees from empirical observations.
>
> [1] Transformers learn in-context by gradient descent
>
> [2] Trained Transformers Learn Linear Models In-Context
>
> [3] Transformers learn to implement preconditioned gradient descent for in-context learning
>
> >Response to W3:
>
> **T1: Ablation on adaptive vs. random feature selection.**
> | Model             | Method   | Australian Credit (validity) | cost        | Budget        | Corporate Rating (validity) | cost        | Budget        |
> |------------------|----------|------------------------------|-------------|---------------|-----------------------------|-------------|---------------|
> | TabPFN-2.5       | random   | 0.82 ± 0.02                 | 3.49 ± 0.27 | 139.14 ± 7.32 | 0.72 ± 0.06                | 3.81 ± 0.13 | 150.00 ± 0.00 |
> |                  | adaptive | 1.00 ± 0.00                 | 3.83 ± 0.04 | 27.01 ± 4.85  | 0.98 ± 0.02                | 4.79 ± 0.19 | 111.71 ± 15.85|
> | Qwen-4B-instruct | random   | 0.45 ± 0.20                 | 3.42 ± 0.03 | 150.00 ± 0.00 | 0.59 ± 0.17                | 3.76 ± 0.06 | 150.00 ± 0.00 |
> |                  | adaptive | 0.79 ± 0.15                 | 3.01 ± 0.06 | 56.01 ± 13.11 | 0.94 ± 0.04                | 3.54 ± 0.06 | 117.32 ± 10.34|
>
> We add an ablation comparing our method with a random subspace baseline (same number of features, same optimizer). As shown in T1, the adaptive strategy consistently improves validity and budget, indicating that the gain comes from adaptive feature selection rather than subspace restriction alone. We will clarify this in the revision.

---

> > ### Author Rebuttal · Reviewer_ETEX · 2026-04-03
> >
> > The authors have resolved my concern, and I am raising my score.

---

### Decision · Program_Chairs · 2026-04-30

**Decision:**

Accept (regular)

**Comment:**

The paper concerns the problem of algorithmic recourse for tabular data under in-context learning. The Authors present a theoretical analysis showing that recourse remains well-defined and bounded under ICL, and characterize how recourse converges toward classical solutions as context size grows. They also propose ASR-ICL, a zeroth-order optimization framework that efficiently generates actionable and sparse recourse for black-box ICL models. The experimental results demonstrate that ASR-ICL achieves recourse quality comparable to classical methods while requiring substantially fewer queries, and empirically confirm, to some extent, the theoretical convergence predictions.

There is broad consensus among the Reviewers that the problem is important and emerging. Nevertheless, they raised concerns including the gap between the theoretical setting (single-layer linear self-attention) and the models evaluated (TabPFN, Qwen, LLaMA, GPT-4o), insufficient justification for why ICL specifically necessitates the proposed method rather than general non-ICL recourse methods, flaws in presentation including inconsistent notation, and missing analysis of computational cost. The Authors addressed most of these concerns during the rebuttal through additional ablations, new baselines, and additional clarifications.

The general and to some extent remaining issues is the question why recourse under ICL is worth studying. As AC I agree with the Authors that this is an interesting direction taking into account the recent successes of ICL-based approaches. Nevertheless, the paper would benefit from clearer articulation of what makes the ICL setting fundamentally different from the classical recourse. Furthermore, the paper would benefit from a more precise delineation of which empirical results are directly supported by the theory versus which are heuristic extensions and from the promised improvements to notation consistency and presentation.